# One Perturbation is Enough: On Generating Universal Adversarial Perturbations against Vision-Language Pre-training Models

## Abstract

Vision-Language Pre-training (VLP) models have exhibited unprecedented capability in many applications by taking full advantage of the multimodal alignment. However, previous studies have shown they are vulnerable to maliciously crafted adversarial samples. Despite recent success, these methods are generally instance-specific and require generating perturbations for each input sample. In this paper, we reveal that VLP models are also vulnerable to the instance-agnostic universal adversarial perturbation (UAP). Specifically, we design a novel Contrastive-training Perturbation Generator with Cross-modal conditions (C-PGC) to achieve the attack. In light that the pivotal multimodal alignment is achieved through the advanced contrastive learning technique, we devise to turn this powerful weapon against themselves, i.e., employ a malicious version of contrastive learning to train the C-PGC based on our carefully crafted positive and negative image-text pairs for essentially destroying the alignment relationship learned by VLP models. Besides, C-PGC fully utilizes the characteristics of Vision-and-Language (V+L) scenarios by incorporating both unimodal and cross-modal information as effective guidance. Extensive experiments show that C-PGC successfully forces adversarial samples to move away from their original area in the VLP model's feature space, thus essentially enhancing attacks across various victim models and V+L tasks.

## 1 Introduction

Vision-Language Pre-training (VLP) models, including ALBEF (Li et al., 2021), TCL (Yang et al., 2022), and BLIP (Li et al., 2022), have recently demonstrated remarkable efficacy in a wide range of Vision-and-Language (V+L) tasks. By self-supervised pre-training on large-scale image-text pairs, VLP models efficiently align cross-modal features and capture rich information from the aligned multimodal embeddings, thereby providing expressive representations for various applications.

Adversarial attacks (Carlini & Wagner, 2017), which aim to deceive models during inference time, have attracted extensive attention due to their significant threat to security-critical scenarios (Eykholt et al., 2018). Recent studies have shown that VLP models are also vulnerable to adversarial samples. The pioneering work Co-Attack (Zhang et al., 2022) proposes the first multimodal attack that simultaneously perturbs both image and text modalities and displays excellent performance. However, Co-Attack only considers relatively easier white-box attacks where victim models are completely accessible. To handle more practical black-box settings, subsequent studies propose various transferable adversarial samples generated on an available surrogate model to fool other inaccessible models. Specifically, SGA (Lu et al., 2023) significantly improves the adversarial transferability through the set-level cross-modal guidance obtained from data augmentations. Subsequently, TMM (Wang et al., 2024) proposes to jointly destroy the modality-consistency features within the clean image-text pairs and include more modality-discrepancy features in the perturbations to further enhance transferability. While existing methods have achieved great success, they are all instance-specific and need to generate a perturbation for each input pair, which results in substantial computational overhead. Meanwhile, universal adversarial attacks, as an efficient instance-agnostic approach that uses only one Universal Adversarial Perturbation (UAP) to conduct attacks, have not been fully investigated for VLP models. This naturally leads to a question, *is it possible to design a universal adversarial perturbation that can effectively deceive VLP models across various image-text pairs?*

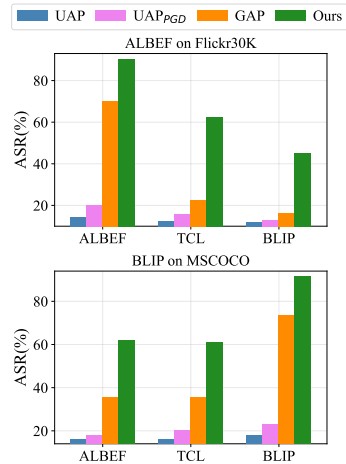

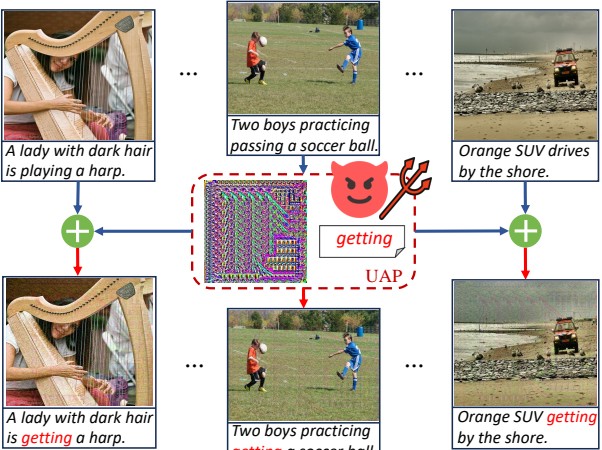

Figure 1: Performance of existing UAP on text retrieval with ALBEF and BLIP as surrogates. UAP$_{PGD}$ indicates the PGD-learned version of UAP.

Figure 2: Illustration of the universal adversarial attacks against VLP models. With only a pair of image-text perturbations, the proposed attack can effectively mislead different models on diverse V+L tasks.

**Motivation.** To this end, we make an intuitive attempt to transplant existing renowned approaches UAP (Moosavi-Dezfooli et al., 2017) and GAP (Poursaeed et al., 2018) to launch attacks on several VLP models by maximizing the distance between the embeddings of the adversarial image and its matched texts. Unfortunately, Figure 1 demonstrates that these methods yield unsatisfactory attack success rates (ASR), especially for black-box attacks. Empirically, this failure stems from their narrow focus on the image modal, disregarding the other modality and the multimodal information that plays a pivotal role in VLP models. To overcome this challenge, we revisit the VLP models' basic training paradigm and emphasize that regardless of the downstream V+L tasks, their achieved outstanding performance is heavily reliant on the well-established multimodal alignment, which draws the embedding of matched image-text pairs closer while distancing those of non-matched pairs. In light of this consideration, we argue that the key core of an effective universal adversarial attack is to obtain a UAP that can fundamentally destroy this learned alignment relationship to mislead VLP models into making incorrect decisions. Besides, Fig. 1 also shows that the generator-based GAP consistently outperforms UAP methods, confirming the superiority of the generative paradigm, which is also corroborated by numerous studies (Gao et al., 2024; Feng et al., 2023).

Based on these insights, we propose a novel generative framework that learns a Contrastive-training Perturbation Generator with Cross-modal conditions (C-PGC) to launch universal attacks on VLP models (see Fig. 2). To essentially destroy the multimodal alignment, we devise to utilize VLP models' most powerful weapons to attack against themselves, i.e., use the contrastive learning mechanism to train the generator using our maliciously constructed image-text pairs that completely violate the correct VL matching relationship, to produce perturbation that pushes the embedding of matched pairs apart while pulling those of non-matched ones together. Inspired by the multimodal characteristics of V+L scenarios, we modify the generator's architecture to incorporate cross-modal knowledge through the advanced cross-attention mechanism for better guidance. In addition, we also consider the intra-modal influence and introduce an unimodal distance loss to further enhance the attacks. Since previous studies (Zhang et al., 2022; Lu et al., 2023) achieve impressive improvements via multimodal perturbation, we are motivated to generate UAP for both images and texts to utilize the synergy between different modalities. Our contributions can be summarized as follows:

- We design a novel cross-modal conditional perturbation generator, which produces effective UAP for both image and text modalities to achieve universal adversarial attacks on VLP models.

- We propose the first malicious contrastive paradigm tailored for multimodal adversarial attacks, which incorporates both unimodal and multimodal guidance to contrastively train the generator using our meticulously constructed positive and negative pairs for enhanced attack effects.

- Extensive experiments on 6 various VLP models across different V+L tasks reveal that our method achieves outstanding white-box performance and black-box transferability in different scenarios.

## 2 RELATED WORK

### 2.1 VISION-LANGUAGE PRE-TRAINING MODELS

VLP models are pre-trained on massive image-text pairs to learn the semantic correlations across modalities and serve diverse multimodal user demands (Chen et al., 2023; Du et al., 2022). We next illustrate the basis of VLP models from multiple perspectives.

**Architectures.** Based on the ways of multimodal fusion, the architectures of VLP models can be classified into two types: *single-stream* and *dual-stream architectures*. Single-stream architectures (Li et al., 2019; Chen et al., 2020) directly concatenate the text and image features, and calculate the attention in the same Transformer block for multimodal fusion. On the contrary, dual-stream architectures (Radford et al., 2021; Li et al., 2022) separately feed the text and image features to different Transformer blocks and leverage the cross-attention mechanism for multimodal fusion.

**Pre-training Objectives.** The pre-training objectives for VLP models mainly include *masked features completion*, *multimodal features matching*, and *specific downstream objectives*. Masked features completion (Chen et al., 2020) encourages VLP models to predict the deliberately masked tokens using the remaining unmasked tokens during pre-training. Multimodal features matching (Li et al., 2021) pre-trains VLP models by learning to precisely predict whether the given image-text pairs are matched. Specific downstream objectives (Anderson et al., 2018) directly utilize the training objectives of downstream tasks (e.g., visual question answering) for pre-training VLP models.

**Downstream Tasks.** In this paper, we mainly consider the following multimodal downstream tasks: (1) Image-text retrieval (ITR) (Wang et al., 2016): finding the most matched image for the given text and vice versa, including image-to-text retrieval (TR) and text-to-image retrieval (IR). (2) Image caption (IC) (Bai & An, 2018): generating the most suitable descriptions for the given image. (3) Visual grounding (VG) (Hong et al., 2019): locating specific regions in the image that correspond with the given textual descriptions. (4) Visual entailment (VE) (Xie et al., 2019): analyzing the input image and text and predicting whether their relationship is entailment, neutral, or contradiction.

### 2.2 ADVERSARIAL ATTACKS

**Instance-specific Attacks on VLP Models.** The adversarial robustness of VLP Models has already become a research focus. Early works (Kim & Ghosh, 2019; Yang et al., 2021) impose perturbations only on single modality and lack cross-modal interactions when attacking multimodal models. To address this issue, Co-Attack (Zhang et al., 2022) conducts the first multimodal white-box attacks on VLP models. On the basis of Co-Attack, Lu et al. (2023) extend the attacks to more rigorous black-box settings and propose SGA, which utilizes set-level alignment-preserving argumentations with carefully designed cross-modal guidance. However, Wang et al. (2024) points out that SGA fails to fully exploit modality correlation, and proposes TMM to better leverage cross-modal interactions by tailoring both the modality-consistency and modality-discrepancy features. Nonetheless, these methods are all instance-specific and need to craft perturbations for each input pair.

**Universal Adversarial Examples.** Universal adversarial attacks (Moosavi-Dezfooli et al., 2017; Mopuri et al., 2018) aim to deceive the victim model by exerting a uniform adversarial modification to all the benign samples. These attacks save the redundant procedures of redesigning perturbations for each input sample and are consequently more efficient than traditional attack strategies. Generally, universal adversarial attacks can be categorized into optimization-based methods (Moosavi-Dezfooli et al., 2017; Wang et al., 2023; Liu et al., 2023) and generation-based methods (Hayes & Danezis, 2018; Gao et al., 2024; Anil et al., 2024). Benefiting from the powerful modeling abilities of generative models, generation-based methods are more versatile and can produce more natural samples than optimization-based ones. In this paper, we explore universal adversarial attacks on VLP models and manage to generate UAP with excellent attack effects and high transferability.

## 3 UNIVERSAL MULTIMODAL ATTACKS

In this section, we first present the problem statement of universal adversarial attacks on VLP models. Next, we introduce the overview of our framework. Finally, we illustrate the detailed design of each proposed technique and summarize the training objective and paradigm of C-PGC.

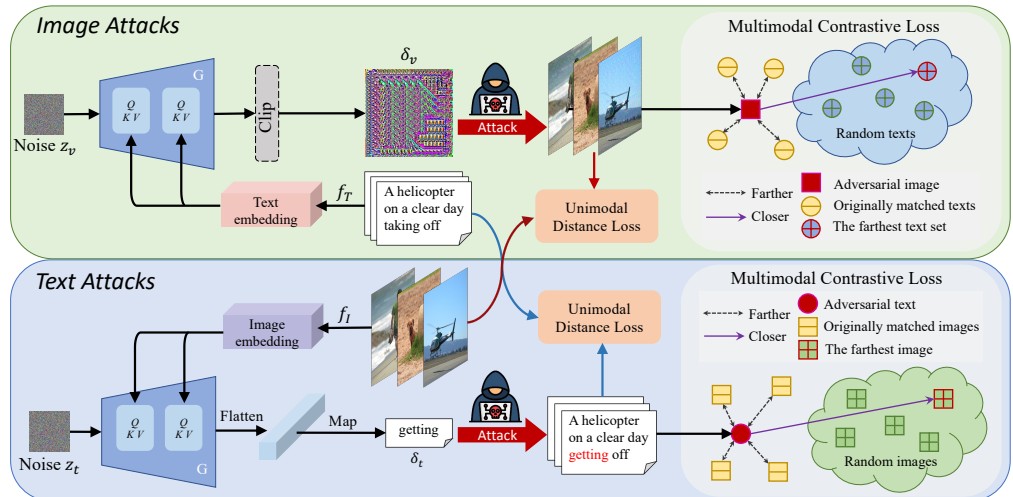

Figure 3: An overview of our proposed universal adversarial attack. Benefiting from the well-designed unimodal distance loss $\mathcal{L}_{Dis}$ and multimodal contrastive loss $\mathcal{L}_{CL}$, the conditional generator learns rich knowledge from features of different modalities and thus produces $\delta_v$ and $\delta_t$ of superior generalization ability across diverse models and downstream tasks.

## 3.1 PROBLEM STATEMENT

We define an input image-text pair as $(v, t)$ and denote $e_v$ and $e_t$ as the image and text embedding encoded by the image encoder $f_I(\cdot)$ and text encoder $f_T(\cdot)$ of the targeted VLP model $f(\cdot)$. Let $\mathcal{D}_s$ be an available dataset consisting of image-text pairs collected by a malicious adversary. The attack objective is to utilize $\mathcal{D}_s$ to train a generator $G_w(\cdot)$ that is capable of producing a powerful pair of universal image-text perturbations $(\delta_v, \delta_t)$ that can affect the vast majority of test dataset $\mathcal{D}_t$ to fool models into making incorrect decisions. Formally, the attack goal can be formulated as:

$$\mathcal{T}(f(v + \delta_v, t \oplus \delta_t)) \neq y, \text{ s.t. } \|\delta_v\|_\infty \leq \epsilon_v, \|\delta_t\|_0 \leq \epsilon_t, \tag{1}$$

where $\mathcal{T}(\cdot)$ denotes the operation that uses the output V+L features to obtain the final predictions, $\oplus$ indicates the text perturbation strategy (Zhang et al., 2022; Lu et al., 2023) that replaces certain important tokens of the original sentence with crafted adversarial words, and $y$ is the correct prediction of the considered V+L task. To ensure the perturbation's imperceptibility, we constrain the pixel-level image perturbation with $l_\infty$ norm of a given budget $\epsilon_v$. The textual perturbation is token-level and the stealthiness is accordingly constrained by the number of modified words $\epsilon_t$. Since altering words in a natural sentence can be easily noticed or detected, we apply a rigorous restriction that permits only a single word to be substituted ($\epsilon_t = 1$). On the premise of imperceptibility, the attacker attempts to generalize the crafted UAP to a wider range of test data and victim models.

## 3.2 OVERVIEW OF THE PROPOSED FRAMEWORK

As depicted in Fig. 3, we adopt the multimodal perturbation strategy and generate perturbations on both image and text modalities for enhanced attacks. Given the similarity between the workflows for image and text, we then take the image attacks as an example to illustrate the proposed framework.

Firstly, a fixed noise $z_v$ is randomly initialized and subsequently fed into the conditional generator. For each image $v$ and its descriptions $\mathbf{t}$, the generator $G_w(\cdot)$ then translates the input noise $z_v$ into the adversarial perturbation $\delta_v$ that is of the same size as the image $v$. During the generation, the network $G_w$ additionally benefits from cross-modal information by integrating the textual embedding, i.e., $\delta_v = G_w(z_v; f_T(\mathbf{t}))$. Next, the generated adversarial noise $\delta_v$ is injected into the clean image to obtain the adversarial image via $v_{adv} = v + \delta_v$. To better guide the training process, we design two efficient unimodal and multimodal losses as our optimization objectives. Unimodal loss is straightforward and aims to push the adversarial images away from the clean images in the latent embedding space, while multimodal loss is based on contrastive learning and uses our manually constructed positive and negative samples to effectively destroy the image-text matching relationship

obtained from feature alignment. Once we finish training the C-PGC using the proposed losses, the input fixed noise is transformed into a UAP that is of great generalization and transferability.

### 3.3 DETAILED DESIGN OF C-PGC

Next, we provide a detailed introduction to each of the proposed designs. Note that we primarily discuss the image attack as an example, given that the design of the text attack is completely symmetrical. The pseudocode of the training procedure is provided in Appendix A.

**Perturbation Generator with Cross-modal conditions.** Previous generative universal attacks (Gao et al., 2024; Anil et al., 2024) have shown excellent efficacy in fooling the discriminative models. Nevertheless, since existing generative attacks are limited to a single modality, directly utilizing the off-the-shelf generators might fail to leverage the multimodal interactions in these special V+L scenarios. To address this limitation, we additionally introduce cross-modal embeddings as auxiliary information to further facilitate the process of perturbation generation. Specifically, we modify the existing generator's architecture by adding several cross-attention modules that have been proven effective in tasks with variable input modalities. The obtained textual embeddings $\boldsymbol{e}_t$ encoded by $f_T(\cdot)$ are then incorporated into our generator through:

$$Q = \boldsymbol{h}_t W_q, K = \boldsymbol{e}_t W_k, V = \boldsymbol{e}_t W_v,$$
$$Attention(Q, K, V) = \text{softmax}\left(\frac{QK^T}{\sqrt{d}}\right) \cdot V, \tag{2}$$

where $\boldsymbol{h}_t \in \mathbb{R}^{B \times d_\alpha}$ is the flattened intermediate features within the generator, $W_q \in \mathbb{R}^{d_\alpha \times d}$, $W_k \in \mathbb{R}^{512 \times d}$, $W_v \in \mathbb{R}^{512 \times d}$ are optimized parameters in the attention modules.

**Multimodal Contrastive Loss.** The preceding analysis regarding the failures of existing UAP attacks encourages us to design a loss function that can guide the generated UAP to break the learned multimodal feature alignment. Motivated by the fact that contrastive learning underpins the cross-modal alignment, we advocate leveraging this mechanism to attack VLP models themselves by contrastively training our C-PGC to essentially disrupt the benign alignment relationship. Concretely, we adopt the widely recognized InfoNCE (He et al., 2020) as our basic contrastive loss.

To establish the contrastive paradigm, we first define the adversarial image $v_{adv}$ as the anchor sample. Besides, it is also necessary to construct an appropriate set of positive and negative samples. Based on the fundamental objective of our attacks, it is natural that the originally matched text descriptions set $\mathbf{t} = \{t_1, t_2 \dots, t_M\}$ can be employed as negative samples $\mathbf{t}_{neg}$ to increase the discrepancy of matched image-text in the feature space of VLP models. To further push the adversarial image $v_{adv}$ away from its correct text descriptions $\mathbf{t}$, we propose a *farthest selection strategy* which utilizes multiple texts whose embeddings differ significantly from that of the original clean image $v$ as positive samples. Specifically, we randomly sample a batch of text sets from $\mathcal{D}_s$ and select the text set with the largest feature distances from the current image $v$ as positive samples, i.e., $\mathbf{t}_{pos} = \{t'_1, t'_2 \dots, t'_K\}$. Moreover, we utilize data augmentations that resize the clean $v$ into diverse scales and apply random Gaussian noise to acquire a more diverse image set $\mathbf{v} = \{v_1, v_2 \dots, v_S\}$ for set-level guidance Lu et al. (2023). With the augmented images and these well-constructed positive and negative samples, the multimodal contrastive loss $\mathcal{L}_{CL}$ can be formulated as:

$$\mathcal{L}_{CL} = log\left(\frac{\sum_{i=1}^{S}\sum_{j=1}^{M} e^{d(v_i+\delta_v, t_j)/\tau}}{\sum_{i=1}^{S}\sum_{j=1}^{M} e^{d(v_i+\delta_v, t_j)/\tau} + \sum_{i=1}^{S}\sum_{j=1}^{K} e^{d(v_i+\delta_v, t'_j)/\tau}}\right), \tag{3}$$

where $\delta_v$ is the universal image perturbation, $\tau$ denotes the temperature parameter and $d(v, t) = Sim(f_I(v), f_T(t))$, where $Sim(\cdot, \cdot)$ represents the cosine similarity measurement.

**Unimodal Distance Loss.** Apart from the multimodal guidance, we also consider the unimodal influence by directly pushing adversarial images away from their initial visual semantic area to further enhance attack effects. Similarly, to acquire set-level guidance, the input image $v$ is initially resized to different scales and then added with Gaussian noise to generate the augmented image set $\mathbf{v} = \{v_1, v_2 \dots, v_S\}$. Then, we craft the adversarial image through $v_{adv} = v + \delta_v$ and process $v_{adv}$ with

the same augmentation operation to obtain the adversarial image set $\mathbf{v}_{adv} = \{v_1^{adv}, v_2^{adv} \ldots, v_S^{adv}\}$. Finally, we minimize the negative Euclidean distance between the embeddings of adversarial images and clean images to optimize the UAP generator. Formally, the loss $\mathcal{L}_{Dis}$ is formulated as:

$$\mathcal{L}_{Dis} = -\sum_{i=1}^{S}\sum_{j=1}^{S}\|f_I(v_i^{adv}) - f_I(v_j)\|_2. \tag{4}$$

Taking advantage of the unimodal set-level guidance, $\mathcal{L}_{Dis}$ ensures an effective optimization direction during the generator training and further improves the attack effectiveness of our UAP.

**Training Objective.** With the two well-designed loss terms $\mathcal{L}_{Dis}$ and $\mathcal{L}_{CL}$, the overall optimization objective of our conditional generator concerning image attacks can be formulated as:

$$\min_{w} \mathbb{E}_{(v,\mathbf{t})\sim\mathcal{D}_s, \mathbf{t}_{pos}\sim\mathcal{D}_s}(\mathcal{L}_{CL} + \lambda\mathcal{L}_{Dis}), \text{ s.t. } \|G_w(z_v; f_T(\mathbf{t}))\|_\infty \le \epsilon_v, \tag{5}$$

where $\lambda$ is the pre-defined hyperparameter to balance the contributions of $\mathcal{L}_{CL}$ and $\mathcal{L}_{Dis}$. By training the network using the proposed loss function based on the data distribution of the multimodal training dataset $D_s$, the generator is optimized to produce UAP that can push the features of mismatched image-text pairs together while pulling the embeddings of the matched ones apart, thereby learning a UAP with excellent generalization ability and adversarial transferability.

**Text Modality Attacks**. In textual attacks, the UAP generator's architecture and training loss are completely symmetrical with those of image attacks. Correspondingly, embeddings of the matched image $v$ are used as the cross-modal conditions for the generator. Given an adversarial text $t_{adv}$ as the anchor sample, we use the set $\mathbf{v} = \{v_1, v_2 \ldots, v_S\}$ scaled from the originally matched image $v$ as negative samples while the $\mathbf{v}' = \{v_1', v_2' \ldots, v_S'\}$ augmented from the farthest image $v'$ within the randomly sampled image set as positive samples to formulate the $\mathcal{L}_{CL}$ loss. $\mathcal{L}_{Dis}$ is consequently calculated as the negative Euclidean distance between the embeddings of $t_{adv}$ and the clean input $t$.

A notable distinction between the image and text attacks is the approach to inject adversarial perturbations. Due to the discreteness of text data, we apply the token-wise substitute strategy (Lu et al., 2023; Wang et al., 2024) that replaces certain important words in the original sentence with crafted adversarial words. Accordingly, the conditional generator is utilized to output the adversarial textual embeddings, which are subsequently mapped back to the vocabulary space to obtain a universally applicable word-level perturbation. Prior to implementing the word replacement, a meticulous process is undertaken to identify the most optimal position within the sentence to insert the perturbation. Our strategy intends to identify and replace the words that are more likely to have a greater influence during the decision-making. Concretely, for each word $w_i$ within a given sentence, we compute the distance between the embeddings of the original sentence and the $w_i$-masked version encoded by the VLP models to determine its contribution. As aforementioned, we set $\epsilon_t = 1$ and choose the single word exerting the highest feature distance as the target for replacement.

## 4 EVALUATION

We first present the experimental setup in Sec. 4.1 and then comprehensively evaluate C-PGC across multiple VLP models in Sec. 4.2. Sec. 4.3 presents results on more downstream V+L tasks to further validate the effectiveness. Besides, sufficient ablation studies in Sec. 4.4 validate the contribution of each proposed technique and explore the impact of several crucial factors. More experiment results such as the cross-domain attacks from Flickr30k to MSCOCO are provided in Appendix E.

### 4.1 EXPERIMENTAL SETUP

**Downstream tasks and datasets.** We conduct a comprehensive study of C-PGC on four downstream V+L tasks, including image-text retrieval (ITR), image captioning (IC), visual grounding (VG), and visual entailment (VE). For ITR tasks, we employ the Flickr30K (Plummer et al., 2015) and MSCOCO (Lin et al., 2014) datasets which are commonly used in previous works (Zhang et al., 2022; Lu et al., 2023). The MSCOCO is also adopted for evaluating the IC task and we test VG and VE tasks on SNLI-VE (Xie et al., 2019) and RefCOCO+ (Yu et al., 2016) respectively.

**Surrogate models and victim models.** We conduct experiments on a wide range of VLP models including ALBEF (Li et al., 2021), TCL (Yang et al., 2022), X-VLM (Zeng et al., 2022), CLIP$_{\text{ViT}}$

Table 1: ASR (%) of our C-PGC and GAP for image-text retrieval tasks on Flickr30k and MSCOCO. TR indicates text retrieval based on the input image, while IR is image retrieval using input text.

| Dataset | Source | Method | ALBEF | | TCL | | X-VLM | | CLIP$_{ViT}$ | | CLIP$_{CNN}$ | | BLIP | |
|---|---|---|---|---|---|---|---|---|---|---|---|---|---|---|
| | | | TR | IR | TR | IR | TR | IR | TR | IR | TR | IR | TR | IR |
| Flickr30k | ALBEF | GAP | 69.78 | 81.59 | 22.15 | 29.97 | 6.61 | 18.37 | 23.4 | 37.54 | 29.92 | 44.29 | 16.09 | 28.12 |
| | | Ours | **90.13** | **88.82** | **62.11** | **64.48** | **20.53** | **39.38** | **43.1** | **65.93** | **54.4** | **72.51** | **44.79** | **56.36** |
| | TCL | GAP | 33.5 | 40.61 | 82.41 | 80.67 | 6.61 | 17.79 | 21.55 | 38.56 | 30.57 | 45.48 | 21.45 | 31.82 |
| | | Ours | **50.26** | **56.29** | **94.93** | **90.64** | **14.94** | **33.96** | **46.92** | **66.41** | **52.98** | **70.66** | **35.75** | **52.52** |
| | X-VLM | GAP | 16.14 | 24.43 | 17.08 | 26.2 | 90.24 | 85.98 | 24.51 | 41.15 | 42.62 | 53.08 | 16.19 | 25.74 |
| | | Ours | **24.46** | **47.77** | **29.19** | **50.15** | **93.29** | **91.9** | **43.47** | **66.03** | **59.2** | **72.79** | **32.39** | **52.24** |
| | CLIP$_{ViT}$ | GAP | 11.72 | 23.34 | 15.32 | 26.39 | 8.54 | 20.48 | 85.73 | 90.45 | 48.83 | 60.78 | 14.83 | 26.46 |
| | | Ours | **23.23** | **38.67** | **25.05** | **41.79** | **15.85** | **35.59** | **88.92** | **93.05** | **66.06** | **75.42** | **26.71** | **45.7** |
| | CLIP$_{CNN}$ | GAP | 13.57 | 25.21 | 19.05 | 28.87 | 11.59 | 23.13 | 27.46 | 43.16 | 73.18 | 81.6 | 15.25 | 27.94 |
| | | Ours | **15.31** | **38.93** | **19.77** | **43.72** | **17.17** | **41.65** | **39.9** | **64.82** | **81.74** | **88.9** | **22.19** | **46.11** |
| | BLIP | GAP | 12.23 | 23.94 | 14.49 | 25.44 | 6.91 | 17.81 | 20.32 | 37 | 26.81 | 43.59 | 47.21 | 73.33 |
| | | Ours | **32.17** | **44.4** | **33.44** | **44.51** | **18.6** | **35.53** | **43.35** | **60.26** | **48.96** | **66.95** | **71.82** | **82.82** |
| MSCOCO | ALBEF | GAP | 82.65 | 84.35 | 53.6 | 45.46 | 15.09 | 15.64 | 25.18 | 29.94 | 28.06 | 35.28 | 37.44 | 33.61 |
| | | Ours | **96.18** | **95.09** | **82.49** | **76.24** | **39.97** | **48.58** | **59.71** | **67.05** | **61.27** | **70.8** | **59.18** | **63.89** |
| | TCL | GAP | 55.92 | 48.22 | 95.16 | 92.29 | 17.34 | 17.01 | 28.73 | 31.19 | 32.27 | 39.81 | 43.59 | 39.64 |
| | | Ours | **76.62** | **71.17** | **96.72** | **93.88** | **42.99** | **48.4** | **70.32** | **79.08** | **74.1** | **82.97** | **62.35** | **66.97** |
| | X-VLM | GAP | 26.35 | 23.72 | 27.8 | 22.91 | 95.1 | 88.84 | 32.39 | 38.16 | 52 | 55.4 | 24.67 | 22.65 |
| | | Ours | **51.46** | **65.71** | **52.8** | **64.99** | **98.89** | **95.79** | **67.42** | **75.45** | **75.49** | **82.58** | **55.74** | **66.7** |
| | CLIP$_{ViT}$ | GAP | 35.96 | 31.91 | 37.33 | 32.56 | 33.42 | 29.25 | 97.71 | 96.04 | 74.63 | 74.67 | 33.47 | 31.99 |
| | | Ours | **46.92** | **53.89** | **46.03** | **50.87** | **41.49** | **48.6** | **98.74** | **98.01** | **81.58** | **86.5** | **47.35** | **57.55** |
| | CLIP$_{CNN}$ | GAP | 28.67 | 27.51 | 29.84 | 27.69 | 26.4 | 24.81 | 39.64 | 40.53 | 90.34 | 91.56 | 24.99 | 26.18 |
| | | Ours | **33.38** | **46.68** | **40.61** | **50.76** | **35.34** | **46.95** | **63.83** | **70.15** | **94.89** | **94.42** | **37.38** | **53.06** |
| | BLIP | GAP | 35.55 | 38.75 | 35.62 | 33.79 | 22.7 | 21.25 | 32.05 | 35.8 | 40.93 | 45.58 | 73.46 | 72.37 |
| | | Ours | **61.95** | **60.92** | **60.95** | **59.57** | **51.81** | **52.53** | **62.23** | **72.51** | **69.61** | **78.44** | **91.67** | **90.42** |

(Radford et al., 2021), CLIP$_{CNN}$ (Radford et al., 2021), and BLIP (Li et al., 2022). Note that for different V+L tasks, we correspondingly select different VLP models for evaluation based on their capability (Wang et al., 2024). For instance, among the six considered VLP models, only ALBEF, TCL, and X-VLM can handle VG tasks, while only ALBEF and TCL can deal with VE tasks.

**Baselines.** To better reveal the superiority of our proposed method in attacking VLP models, we transplant a representative and powerful algorithm GAP (Poursaeed et al., 2018) to the multimodal attack scenarios by appropriately modifying its original loss function (Lu et al., 2023).

**Implementation details.** Following (Lu et al., 2023), we adopt Karpathy split (Karpathy & Fei-Fei, 2015) to preprocess the dataset and build the test set for evaluation. The test set is disjoint with the generator's training data for rigorous assessment. To ensure the perturbation invisibility, we follow (Wang et al., 2024) and limit the perturbation budgets $\epsilon_v$ to $12/255$ and $\epsilon_t$ to $1$. During the augmentation, we resize the original images into five scales $\{0.5, 0.75, 1, 1.25, 1.5\}$, and apply Gaussian noise with a mean of $0$ and a standard deviation of $0.5$. See Appendix G for more details.

## 4.2 UNIVERSAL ATTACK EFFECTIVENESS

To align with previous studies (Zhang et al., 2022; Lu et al., 2023), we first consider the typical V+L task image-text retrieval and calculate the ASR as the proportion of successful adversarial samples within the originally correctly predicted pairs based on R@1 retrieval results. Appendix E provides results of R@5 and R@10. Experimental results across six 4.2 VLP models are presented in Table 1. We also provide the visualization of the image retrieval on the MSCOCO dataset in Figure 4.

**White-box attack performance.** By observing the white-box ASR in the gray-shaded area, we demonstrate that the proposed algorithm stably achieves excellent ASR on all the evaluated VLP models, validating the outstanding capability of the produced UAP. With only a single pair of per-turbations, we reach a noteworthy average white-box ASR of over 90% on two large datasets in terms of both TR and IR tasks. Especially on the MSCOCO dataset, our method achieves over 95% average ASR on ITR tasks across six surrogate models. Compared with the GAP, the proposed

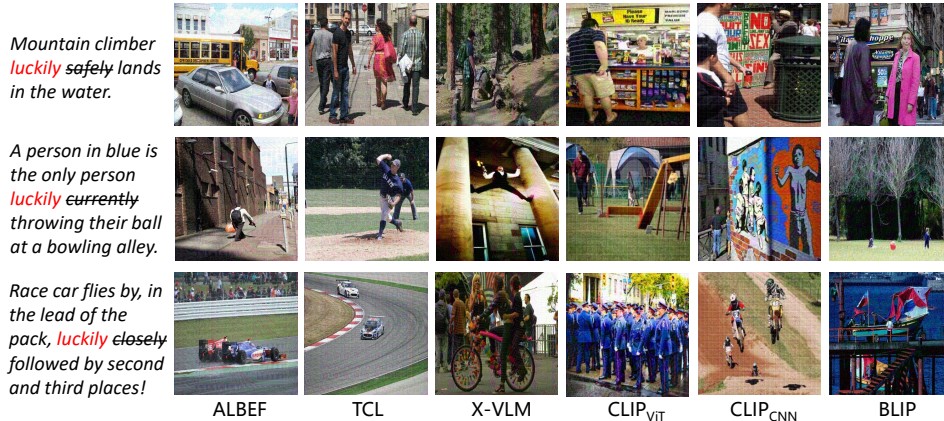

Figure 4: Illustration of image retrieval. The red indicates the universal adversarial word and the crossed-out word is the replaced one. We generate the UAP on ALBEF and test it on 6 target models. All retrieved images do not accurately correspond to the query text, validating the design of C-PGC.

method significantly improves the average fooling rates by nearly 10%, confirming the great validity of our suggested multimodal contrastive-learning mechanism. Essentially, the exceptional performance stems from the efficacy of our generated UAP in destroying the alignment between the image and text modalities, thereby misleading the VLP model during inference.

**Black-box attack performance**. We also conduct thorough experiments regarding the adversarial transferability of the generated UAP by transferring from surrogate models to other inaccessible models. As demonstrated in Table 1, the proposed C-PGC displays great attack performance in the more realistic black-box scenarios, e.g., 82.97% from TCL to CLIP_CNN on MSCOCO for IR tasks. We highlight that the advantage of C-PGC over GAP (Poursaeed et al., 2018) is greatly amplified in the more challenging black-box scenarios, which achieves a significant average improvement of 18.36% and 26.32% for Flickr30K and MSCOCO respectively. These experimental results indicate that our generative contrastive learning framework does not overly rely on the encoded feature space tailored to the surrogate model. Conversely, it is well capable of transferring to breaking the multi-modal alignment of other unseen target models, thus attaining superior adversarial transferability.

Table 2: ASR (%) of ITR tasks under defense strategies. Surrogate model is ALBEF and the dataset is Flick30K. LT denotes the LanguageTool that corrects adversarial words within the sentence.

| Method | ALBEF | | TCL | | X-VLM | | CLIP_ViT | | CLIP_CNN | | BLIP | |
|---|---|---|---|---|---|---|---|---|---|---|---|---|
| | TR | IR | TR | IR | TR | IR | TR | IR | TR | IR | TR | IR |
| Gaussian | 37.92 | 49.49 | 32.4 | 47.04 | 19.31 | 37.79 | 42.49 | 65.61 | 50 | **72.23** | 29.65 | 48.77 |
| Medium | 53.13 | 61.6 | 39.54 | 51.96 | 20.43 | 39.69 | 46.31 | 66.92 | 57.9 | 74.51 | 33.75 | 52.68 |
| Average | 29.09 | 44.91 | 29.61 | 44.72 | 17.89 | 36.07 | 42.98 | **65.42** | **49.74** | 72.48 | **27.55** | **46.9** |
| JPEG | 59.3 | 63.7 | 42.34 | 52.52 | 21.65 | 41.58 | **41.26** | 65.77 | 53.5 | 72.62 | 37.01 | 55.04 |
| DiffPure | 64.34 | 74.63 | 65.22 | 74.8 | 66.06 | 75.19 | 78.08 | 86.7 | 82.25 | 88.03 | 70.45 | 79.09 |
| NRP | 32.33 | 40.63 | **20.19** | 39.23 | **14.63** | 32.62 | 48.4 | 69 | 59.72 | 74.09 | 30.28 | 52.2 |
| NRP+LT | **29.05** | **35.23** | 21.33 | **37.41** | 15.55 | **29.63** | 47.19 | 67.35 | 56.82 | 73.47 | 28.23 | 50.59 |

**Defense Strategies.** We next analyze several defense strategies to mitigate the potential harm brought by the proposed C-PGC. Specifically, we totally align with TMM (Wang et al., 2024) and consider several input preprocessing-based schemes, including image smoothing (Ding et al., 2019) (Gaussian, medium, average smoothing), JPEG compression (Dziugaite et al., 2016), NRP (Naseer et al., 2020), and the prevalent DiffPure (Nie et al., 2022), a powerful purification defense using diffusion models. For adversarial text correction, we choose the LanguageTool (LT) (Wang et al., 2024), which has been widely adopted in various scenarios due to its universality and effectiveness.

The attack results in Table 2 demonstrate that the proposed attack still attains great ASR against different powerful defenses. It also indicates that NRP+LT would be a decent choice to alleviate the threat brought by C-PGC. Another noteworthy finding is that, although DiffPure (Nie et al., 2022) exhibits remarkable performance in defending attacks in classification tasks, its ability is greatly

reduced in V+L scenarios since the denoise process could also diminish some texture or semantic information that is critical for VLP models, thereby acquiring unsatisfactory defense effects.

## 4.3 EVALUATION ON MORE DOWNSTREAM TASKS

We further demonstrate C-PGC's ability to destroy the multimodal alignment by presenting more results on diverse V+L tasks. Specifically, we consider Image Captioning (IC), Visual Grounding (VG), and Visual Entailment (VE). The results of VE are shown in Appendix E due to space limit.

Table 3: Attacks results of image captioning. The Baseline represents the performance of the target model on clean data. The used dataset is MSCOCO.

| Source | B@4 | METEOR | ROUBE_L | CIDEr | SPICE |
|---|---|---|---|---|---|
| Baseline | 39.7 | 31.0 | 60.0 | 133.3 | 23.8 |
| ALBEF | 30.1 | 23.7 | 51.2 | 92.5 | 17.5 |
| TCL | 29.5 | 23.5 | 51.0 | 88.9 | 17.3 |
| BLIP | **21.2** | **19.1** | **45.5** | **62.5** | **13.7** |

**Image captioning.** The objective of IC is to generate text descriptions relevant to the semantic content based on the given image. We use ALBEF, TCL, and BLIP as source models and attack the commonly used captioning model BLIP. Similar to SGA (Lu et al., 2023), several typical evaluation metrics of IC are calculated to measure the quality of generated captions, including BLEU (Papineni et al., 2002), METEOR (Banerjee & Lavie, 2005), ROUGE (Lin, 2004), CIDEr (Vedantam et al., 2015), and SPICE (Anderson et al., 2016). The results in Table 3 demonstrate that our algorithm again displays prominent attack effects, e.g., the crated UAP induces notable drops of 10.2% and 9% in the B@4 and ROUGE_L respectively when transferred from TCL to BLIP.

Table 4: Attack results of visual grounding. The first row displays the source models, where the Baseline indicates the clean performance of the target model on clean data.

| Target | Baseline | | | ALBEF | | | TCL | | | X-VLM | | |
|---|---|---|---|---|---|---|---|---|---|---|---|---|
| | Val | TestA | TestB | Val | TestA | TestB | Val | TestA | TestB | Val | TestA | TestB |
| ALBEF | 58.4 | 65.9 | 46.2 | **37.1** | **39.8** | **32.0** | 42.2 | 46.9 | 35.2 | 37.6 | 40.2 | 33.0 |
| TCL | 59.6 | 66.8 | 48.1 | 43.6 | 47.8 | 36.9 | **39.0** | **41.4** | **33.6** | 39.5 | 41.7 | 34.1 |
| X-VLM | 70.8 | 67.8 | 61.8 | 51.8 | 54.7 | 47.7 | 52.7 | 55.9 | 47.8 | **33.1** | **34.7** | **28.8** |

**Visual grounding.** This is another common V+L task, which aims to locate the correct position in an image based on a given textual description. We conduct experiments on RefCOCO+ using ALBEF, TCL, and X-VLM as source and target models. Table 4 indicates that C-PGC brings a notable negative impact on the localization accuracy in both white-box and black-box settings, again verifying that the produced UAP strongly breaks the cross-modal interaction and alignment.

## 4.4 ABLATION STUDY

This subsection employs the representative ALBEF (Li et al., 2021) model as the surrogate model to provide sufficient ablation studies on Flickr30K. We begin our analysis on the contribution of each proposed technique. Subsequently, we examine the sensitivity of certain hyperparameters.

**The effect of $\mathcal{L}_{CL}$ and $\mathcal{L}_{Dis}$.** To investigate the impact of the proposed loss terms, we introduce two variants C-PGC$_{CL}$ and C-PGC$_{Dis}$ that remove $\mathcal{L}_{CL}$ and $\mathcal{L}_{Dis}$ from the training loss respectively. As shown in Table 5, the removal of $\mathcal{L}_{CL}$ leads to significant degradation, particularly for black-box transferable attacks. e.g., a 27.12% ASR drop in TR tasks from ALBEF to TCL. This validates the considerable contribution of $\mathcal{L}_{CL}$ to guarantee a successful attack. Regarding the influence of $\mathcal{L}_{Dis}$, we demonstrate that this unimodal guidance can further enhance attacks on the basis of $\mathcal{L}_{CL}$, e.g., a 10.59% increase in the ASR of TR tasks for white-box attacks on ALBEF. The proposed two loss terms complement each other and jointly underpin the generalizability of the generated UAP.

**The effect of positive sample selection.** To validate the farthest selection strategy when constructing positive samples, we design another variant C-PGC$_{Rand}$ that adopts randomly sampled data points as positive samples. Results in Table 5 reveal the necessity of the proposed farthest selection strategy as it brings an average improvement of 25.96% in white-box ASR and 4.95% in black-box ASR. Moreover, we can also conclude that if the positive samples are not adequately defined, adding $\mathcal{L}_{CL}$ would even severely harm the white-box performance (see C-PGC$_{CL}$ and C-PGC$_{Rand}$).

Table 5: ASR (%) of C-PGC and its variants averaged across six target models on retrieval tasks.

| Method | ALBEF | | TCL | | X-VLM | | CLIP$_{ViT}$ | | CLIP$_{CNN}$ | | BLIP | |
|---|---|---|---|---|---|---|---|---|---|---|---|---|
| | TR | IR | TR | IR | TR | IR | TR | IR | TR | IR | TR | IR |
| C-PGC | **90.13** | **88.82** | **62.11** | **64.48** | **20.53** | **39.38** | **43.1** | **65.93** | **54.4** | **72.51** | **44.79** | **56.36** |
| C-PGC$_{CL}$ | 76.46 | 77.58 | 34.99 | 47.55 | 14.33 | 33.61 | 42.98 | 62.81 | 46.11 | 65.58 | 27.13 | 46.44 |
| C-PGC$_{Dis}$ | 79.54 | 82.46 | 56.52 | 62.21 | 20.24 | 38.26 | 39.78 | 65.1 | 52.2 | 71.01 | 42.43 | 55.52 |
| C-PGC$_{Rand}$ | 61.87 | 65.17 | 43.69 | 52.54 | 19.51 | 35.47 | 40.33 | 65.77 | 54.15 | 70.62 | 39.43 | 52.59 |
| C-PGC$_{CA}$ | 85.18 | 83.07 | 45.76 | 53.73 | 15.24 | 34.02 | 39.29 | 60.61 | 47.15 | 40.64 | 32.39 | 48.29 |

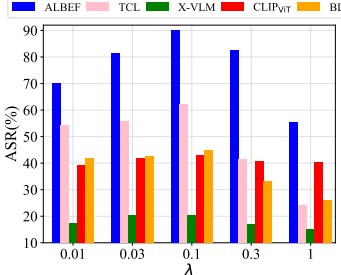 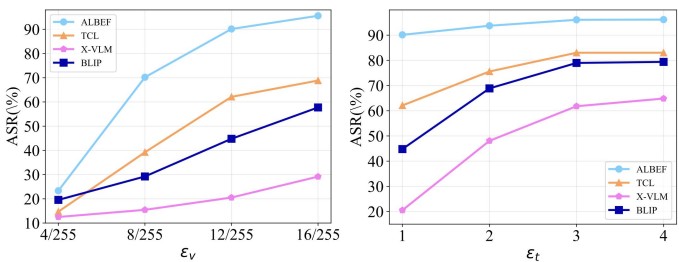

Figure 5: ASR of five target models on TR tasks under various $\lambda$.

Figure 6: ASR of five target models on the TR task under different values of perturbation budgets $\epsilon_v$ and $\epsilon_t$ respectively.

**The effect of cross-modal conditions.** As aforementioned, cross-attention (CA) modules are introduced into the generator to exploit cross-modal information. We then design C-PGC$_{CA}$ that cancels these CA layers to explore their influence. As expected, it causes a notable 9.78% average decrease across six target models, confirming the vital role of cross-modal knowledge. An interesting finding is that C-PGC$_{CA}$ induces a more pronounced drop in black-box attacks than white-box ones, indicating that cross-modal conditions exert a greater contribution to the adversarial transferability.

**Different regulatory factor $\lambda$.** The value of $\lambda$ is a critical factor as it adjusts the scales of the two loss terms $\mathcal{L}_{CL}$ and $\mathcal{L}_{Dis}$. We explore the attack performance under various values of $\lambda$ to confirm the optimal value. Figure 5 indicates that $\lambda = 0.1$ achieves superior performance.

**Different perturbation budgets $\epsilon_v$ and $\epsilon_t$.** As shown in Figure 6, we analyze varying perturbation budgets for $\epsilon_v$ and $\epsilon_t$. Generally, the ASR increases with the larger perturbation magnitudes. Note that when $\epsilon_v = 4/255$, C-PGC's performance is severely compromised since the budget $4/255$ is too small to allow the UAP to carry enough information required to generalize to diverse data samples. We also find that the improvement slows down as $\epsilon_v$ increases from $12/255$ to $16/255$. Thus, we select the moderate value of $12/255$ to reach a balance between attack utility and imperceptibility. For text perturbation, $\epsilon_t$ exhibits a more profound influence on the black-box attacks. In our experiments, we strictly set $\epsilon_t = 1$ for invisibility. However, attackers can adjust the value of $\epsilon_t$ in accordance with their demands to trade off the attack efficacy and the perturbation stealthiness.

## 5 CONCLUSION

This paper delves into the challenging task of launching universal adversarial attacks against VLP models and proposes an effective solution that achieves superior performance using only one universal pair of image-text perturbations. We begin by revealing the unsatisfactory results of existing UAP methods and empirically explaining the underlying reasons. Based on our analysis, we propose to break the crucial cross-modal alignment in VLP models by designing a contrastive-learning generative UAP framework that leverages both unimodal and multimodal information to enhance the attacks. Extensive experiments validate the efficacy of the proposed algorithm on diverse VLP models and V+L tasks. We highlight that the proposed framework makes a significant step in exploring the classic universal adversarial attacks in VLP models and deepens our understanding of the mechanism of VLP models. We also hope that this paper can promote future research that explores more sophisticated defenses to strengthen the resilience of VLP models against adversarial attacks.

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

## A    PSEUDOCODE OF THE PROPOSED ALGORITHM

We present the pseudocode of our proposed attack algorithm for image modality in Alg. 1. Note that the text attacks are completely symmetrical as illustrated in Sec. 3.3.

---

**Algorithm 1** Pseudocode of universal image attacks

---

**Require:** $G_w(\cdot)$: the perturbation generator; $D_s$: the multimodal training set; $f_I, f_T$: image encoder and text encoder of the surrogate VLP model; $N$: the max iteration; $\epsilon_v$: the perturbation budget; $S$: the scaling times;
**Ensure:** Universal image perturbation $\delta_v$;
 1: **Initialize** the fixed noise $z_v$ with Gaussian distribution;
 2: **for** $i \leftarrow 0$ to $N$ **do**
 3:    Randomly sample an image-text pair $(v, \mathbf{t}) \sim D_s$;
 4:    $\delta_v = Clip_{\epsilon_v}(G_w(z_v; f_T(\mathbf{t}))), v_{adv} = v + \delta_v$;
 5:    Augment $v$ and $v_{adv}$ into different scales and apply random Gaussian noises to obtain $\mathbf{v} = \{v_1 \ldots, v_S\}$ and $\mathbf{v}_{adv} = \{v_1^{adv} \ldots, v_S^{adv}\}$;
 6:    Randomly sample a batch of text sets from $D_s$ and obtain $\mathbf{t}_{pos} = \{t_1' \ldots, t_K'\}$ by selecting the one with the farthest feature distance from the clean image $v$;
 7:    Compute $\mathcal{L}_{CL}$ with $\mathbf{v}_{adv}$, $\mathbf{t}$ and $\mathbf{t}_{pos}$ by Eq. (3);
 8:    Compute $\mathcal{L}_{Dis}$ with $\mathbf{v}$ and $\mathbf{v}_{adv}$ by Eq. (4);
 9:    Optimize the generator $G_w$ based on Eq. (5);
10:    Backward pass and update $G_w$;
11: **end for**
12: **Return** $\delta_v$

---

## B    RATIONAL BEHIND OUR DESIGN OF LOSS FUNCTION

It is widely acknowledged that contrastive learning serves as a powerful and foundational tool for modality alignment in VLP models, establishing a nearly point-to-point relationship between image and text features. Our core idea stems from the general principle: *"It's easier to tear down than to build up."* Since contrastive learning can establish robust and precise alignment, leveraging the same technique to disrupt the established alignments is expected to yield effective performance.

Taking image attack as an example, the principle behind our contrastive learning-based attack can be understood from two perspectives:

• Leverage the originally matched texts as negative samples to push the aligned image-text pair apart. This broadly corresponds to the common objective of untargeted attacks.

• Additionally, our contrastive paradigm introduces dissimilar texts as positive samples to pull the adversarial image out of its original subspace and relocate it to an incorrect feature area.

By simultaneously harnessing the collaborative effects of *push* (negative samples) and *pull* (positive samples), the proposed contrastive framework achieves exceptional attack performance, which has been validated by comprehensive experimental results.

Table 6: ASR results of the proposed method with different loss functions on Flickr30 when the surrogate model is ALBEF.

| Method | ALBEF | | TCL | | X-VLM | | CLIP$_{ViT}$ | | CLIP$_{CNN}$ | | BLIP | |
|---|---|---|---|---|---|---|---|---|---|---|---|---|
| | TR | IR | TR | IR | TR | IR | TR | IR | TR | IR | TR | IR |
| $\mathcal{L}_{MSE}$ | 12.02 | 30.75 | 14.39 | 35.08 | 11.41 | 30.79 | 37.32 | 56.05 | 40.17 | 56.39 | 19.66 | 37.33 |
| $\mathcal{L}_{Cos}$ | 57.55 | 67.4 | 37.06 | 49.45 | 10.7 | 28.48 | 37.49 | 58.3 | 40.87 | 58.39 | 23.33 | 39.44 |
| $\mathcal{L}_{CL}$ | **76.46** | **82.46** | **56.52** | **62.61** | **14.33** | **33.61** | **42.98** | **62.81** | **46.11** | **65.58** | **27.13** | **46.44** |
| $\mathcal{L}_{MSE}$+$\mathcal{L}_{Dis}$ | 81.09 | 83.71 | 48.76 | 56.54 | 17.58 | 35.72 | 41.5 | 64.72 | 47.41 | 70.34 | 35.96 | 51.76 |
| $\mathcal{L}_{Cos}$+$\mathcal{L}_{Dis}$ | 65.20 | 72.71 | 36.13 | 50.06 | 18.63 | 36.74 | 42.23 | 65.17 | 50.91 | 69.78 | 36.91 | 50.69 |
| $\mathcal{L}_{CL}$+$\mathcal{L}_{Dis}$ | **90.13** | **88.82** | **62.11** | **64.48** | **20.53** | **39.38** | **43.1** | **65.93** | **54.4** | **72.51** | **44.79** | **56.36** |

Besides, we also explore several potential alternative loss functions that more directly align with the common untargeted attack Table 6, including maximizing the negative cosine similarity $\mathcal{L}_{Cos}$ or MSE distance $\mathcal{L}_{MSE}$ between the features of matched image-text pairs.

Recall that $\mathcal{L}_{CL}$ and $\mathcal{L}_{Dis}$ denote the proposed contrastive loss and the unimodal loss term respectively. As observed, the use of $\mathcal{L}_{CL}$ consistently brings significant ASR improvements, verifying the rationality and superiority of contrastive loss.

## C  COMPARISON WITH A CONCURRENT STUDY

We notice a concurrent study (Zhang et al., 2024) on UAP attacks for VLP models, which also shows promising attack performance. To make a fair comparison, we faithfully reproduce this algorithm using their publicly released code under the same experimental settings as ours. Note that Zhang et al. (2024) implement several versions of their method and we report their best results in Table 7.

Table 7: Comparison of C-PGC with a recent attack (Zhang et al., 2024) on Flicke30K

| Source | Method | ALBEF | | TCL | | X-VLM | | CLIP$_{ViT}$ | | CLIP$_{CNN}$ | | BLIP | |
|---|---|---|---|---|---|---|---|---|---|---|---|---|---|
| | | TR | IR | TR | IR | TR | IR | TR | IR | TR | IR | TR | IR |
| ALBEF | ETU | 78.01 | 84.56 | 29.92 | 35.91 | 14.33 | 22.03 | 23.77 | 39.2 | 33.55 | 47.69 | 22.61 | 32.28 |
| | C-PGC | **90.13** | **88.82** | **62.11** | **64.48** | **20.53** | **39.38** | **43.1** | **65.93** | **54.4** | **72.51** | **44.79** | **56.36** |
| CLIP$_{ViT}$ | ETU | 14.8 | 25.23 | 21.22 | 30.87 | 10.87 | 24.96 | 84.14 | 90.45 | 57.51 | 65.51 | 16.4 | 27.22 |
| | C-PGC | **23.23** | **38.67** | **25.05** | **41.79** | **15.85** | **35.59** | **88.92** | **93.05** | **66.06** | **75.42** | **26.71** | **45.7** |

By contrastively training the conditional generator, the proposed C-PGC greatly enhances the attack by achieving significant improvements in ASR. Particularly in the more realistic and challenging transferable scenarios, the proposed method achieves considerably better performance, e.g., 32.19% and 28.57% increase in ASR of TR and IR tasks when transferring from ALBEF to TCL. These results strongly confirm the superiority of our contrastive learning-based generative paradigm.

## D  SEMANTIC SIMILARITY ANALYSIS

The basic objective of untargeted adversarial attacks is to fool the victim model to output incorrect predictions (Dong et al., 2018), while the attacker is supposed to preserve semantic similarity between the original and the adversarial sample to ensure attack imperceptibility. In our implementation, we follow the rigorous setup in prior works (Zhang et al., 2022; Lu et al., 2023; Wang et al., 2024) that modify only one single word to preserve semantic similarity and attack stealthiness. To quantitatively analyze the semantic similarity, we provide the BERT scores (Zhang et al.), which calculate the P (precision), R (recall), and F1 (F1 score) as results for the semantic distance between 5,000 clean and adversarial sentences in Table 8.

Table 8: Comparison of BERTScore between clean and adversarial texts.

| Method | ALBEF | | | TCL | | | CLIP$_{ViT}$ | | | CLIP$_{CNN}$ | | |
|---|---|---|---|---|---|---|---|---|---|---|---|---|
| | P | R | F1 | P | R | F1 | P | R | F1 | P | R | F1 |
| Co-Attack | 0.8328 | 0.8589 | 0.8455 | 0.8325 | 0.8588 | 0.8453 | 0.8269 | 0.8526 | 0.8394 | 0.8271 | 0.853 | 0.8397 |
| SGA | 0.8389 | **0.8654** | 0.8518 | 0.8376 | 0.8646 | 0.8509 | 0.8416 | **0.8697** | 0.8553 | 0.8378 | 0.865 | 0.8511 |
| Ours | **0.8891** | 0.8613 | **0.8748** | **0.8924** | **0.8687** | **0.8802** | **0.8746** | 0.8684 | **0.8713** | **0.8948** | **0.8842** | **0.8893** |

Note that we provide previous sample-specific algorithms Co-Attack (Zhang et al., 2022) and SGA (Lu et al., 2023) as references. Notably, our method achieves better similarity scores to these wide-acknowledged sample-specific methods across different surrogate VLP models, demonstrating the outstanding attack stealthiness of our C-PGC. The lower semantic similarity of sample-specific methods essentially stems from their word-selection mechanism, which maximizes the semantic distance tailored to every input sentence for attack enhancement. Specifically, to achieve better performance, these methods select the adversarial word that maximizes the distance between the original and perturbed text for every input sentence, which inherently leads to relatively larger semantic

deviations. This highlights that our universal attack achieves a better balance between efficacy and stealthiness. Besides, we also provide BLEU metrics when the surrogate model is ALBEF in Table 9. These results again validate the better stealthiness of our C-PGC.

Table 9: Comparison of BLEU metrics between clean and adversarial texts.

| Method | B@4 | METEOR | ROUBE_L | CIDEr | SPICE |
|---|---|---|---|---|---|
| Co-Attack | 0.79 | 0.52 | 0.895 | 7.03 | 0.661 |
| SGA | 0.798 | 0.527 | 0.898 | 7.159 | 0.668 |
| Ours | **0.889** | **0.552** | **0.905** | **8.036** | **0.671** |

## E    MORE EXPERIMENTAL RESULTS

In this section, we provide more experimental results of our method in various tasks and scenarios.

Table 10: ASR results of C-PGC and C-PGC$_{Sin}$ with a single text as the positive sample.

| Source | Method | ALBEF | | TCL | | X-VLM | | CLIP$_{ViT}$ | | CLIP$_{CNN}$ | | BLIP | |
|---|---|---|---|---|---|---|---|---|---|---|---|---|---|
| | | TR | IR | TR | IR | TR | IR | TR | IR | TR | IR | TR | IR |
| ALBEF | C-PGC$_{Sin}$ | 82.99 | 86.14 | 49 | 56.98 | 18.19 | 35.79 | 40.52 | 65.9 | 51.09 | 69.68 | 38.54 | 52.86 |
| | C-PGC | **90.13** | **88.82** | **62.11** | **64.48** | **20.53** | **39.38** | **43.1** | **65.93** | **54.4** | **72.51** | **44.79** | **56.36** |
| CLIP$_{ViT}$ | C-PGC$_{Sin}$ | 20.55 | 37.46 | 24.43 | 41.39 | 13.52 | 32.6 | 79.93 | 88.64 | 55.44 | 69.43 | 24.4 | 43.06 |
| | C-PGC | **23.23** | **38.67** | **25.05** | **41.79** | **15.85** | **35.59** | **88.92** | **93.05** | **66.06** | **75.42** | **26.71** | **45.7** |

**Diverse target texts as positive samples.** We first investigate the effects of using multiple targets for contrastive training in maximizing the distance between adversarial and original images. We implement a variant C-PGC$_{Sin}$, which uses only a single target text with the farthest distance as the positive sample. The results in Table 10 illustrate that the use of multiple target texts can enhance attack effectiveness, validating the efficacy of set-level diverse guidance.

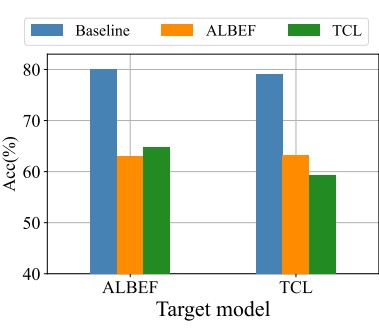

Figure 7: Accuracy of VE tasks for different source and target models.

**Visual entailment tasks.** Given an image and a textual description, visual entailment involves determining whether the textual description can be inferred from the semantic information of the image. We align with previous VLP attacks (Zhang et al., 2022; Wang et al., 2024) and conduct experiments on the SNLI-VE (Xie et al., 2019) dataset using the ALBEF and TCL models. Note that the Baseline represents the clean performance of the target model on the clean data and the orange and green indicate ALBEF and TCL as source models respectively. The results presented in Figure 7 reveal that C-PGC obtains impressive attack effects by decreasing the average accuracy by nearly 20%. Notably, (Do et al., 2020) has reported a large number of annotation errors in the labels of the SNLI-VE corpus used for VE tasks. Therefore, the presented results are only for experimental integrity and reference purposes.

**Ablation study of the data augmentation.** As in the main text, we are motivated by the significant gains introduced by SGA's augmentation Lu et al. (2023) and hence integrate it into the proposed framework to enhance the universal perturbation. The underlying mechanism is to leverage the many-to-many relationships between images and texts by introducing multiple augmented images to provide diverse guidance and improve the optimization direction.

To reveal its effectiveness and explore alternative augmentation techniques, we devise three variants, including C-PGC$_{NoAug}$ without any augmentation, C-PGC$_{ScMix}$ and C-PGC$_{Admix}$ with the ScMix Zhang et al. (2024) and Admix Wang et al. (2021) respectively. The results are shown in Table 11.

It can be observed that the set-level augmentation brings significant improvements over the no augmentation baseline and C-PGC with the current augmentation strategy outperforms the ScMix

Table 11: Attack performance under different data augmentation strategies using Flickr30K.

| Source | Method | ALBEF | | TCL | | X-VLM | | CLIP$_{ViT}$ | | CLIP$_{CNN}$ | | BLIP | |
|---|---|---|---|---|---|---|---|---|---|---|---|---|---|
| | | TR | IR | TR | IR | TR | IR | TR | IR | TR | IR | TR | IR |
| ALBEF | C-PGC$_{ScMix}$ | 66.08 | 76.26 | 39.03 | 51.24 | 20.73 | 37.47 | 40.02 | 65.58 | 50.13 | 71.85 | 34.6 | 51.9 |
| | C-PGC$_{Admix}$ | 62.8 | 72.23 | 34.47 | 47.78 | 19 | 36.67 | 42 | 64.88 | 48.19 | 69.68 | 32.28 | 50.05 |
| | C-PGC$_{NoAug}$ | 69.78 | 74.79 | 47 | 57.26 | 20.43 | 37.55 | 42.36 | 65.17 | 53.63 | 71.6 | 41.22 | 55.34 |
| | C-PGC | **90.13** | **88.82** | **62.11** | **64.48** | **20.53** | **39.38** | **43.1** | **65.93** | **54.4** | **72.51** | **44.79** | **56.36** |
| CLIP$_{ViT}$ | C-PGC$_{ScMix}$ | 20.55 | 37.46 | 24.43 | 41.39 | 13.52 | 32.6 | 79.93 | 88.64 | 55.44 | 69.43 | 24.4 | 43.06 |
| | C-PGC$_{Admix}$ | 19.53 | 37.04 | 24.02 | 41.5 | 14.74 | 34.26 | 85.34 | 91.8 | 59.07 | 71.78 | 23.66 | 43.22 |
| | C-PGC$_{NoAug}$ | 18.5 | 37.8 | 22.19 | 39.86 | 13.47 | 34.17 | 86.46 | 87.11 | 61.53 | 71.36 | 25.03 | 44.73 |
| | C-PGC | **23.23** | **38.67** | **25.05** | **41.79** | **15.85** | **35.59** | **88.92** | **93.05** | **66.06** | **75.42** | **26.71** | **45.7** |

(Zhang et al., 2024) and Admix (Wang et al., 2021), revealing that the set-level guidance is more suitable for our contrastive training. This is achieved by SGA's alignment-preserving augmentation, which enriches image-text pairs while maintaining their inherent alignments intact Lu et al. (2023).

**Cross-domain scenarios.** We proceed to discuss the attack performance of the proposed algorithm in a more challenging scenario where there is an obvious distribution shift between the training dataset and the test samples. Specifically, we generate universal adversarial perturbations based on MSCOCO or Flickr30K and evaluate them accordingly on the other dataset. We present the attack success rates on the retrieval tasks across six models in Table 12. It can be observed that the domain gap indeed has a negative effect on attack performance. However, our method still maintains excellent ASR in most cases, unveiling its outstanding cross-domain transferability.

Table 12: ASR (%) of Cross-domain attacks on six models from Flickr30k to MSCOCO and vice versa. The gray shading indicates white-box attacks.

| Setting | Source | ALBEF | | TCL | | X-VLM | | CLIP$_{ViT}$ | | CLIP$_{CNN}$ | | BLIP | |
|---|---|---|---|---|---|---|---|---|---|---|---|---|---|
| | | TR | IR | TR | IR | TR | IR | TR | IR | TR | IR | TR | IR |
| Flickr30K → MSCOCO | ALBEF | **96.83** | **94.69** | 81.46 | 74.87 | 44.79 | 51.64 | 63.68 | 73.06 | 69.77 | 78.09 | 68.88 | 70.61 |
| | TCL | 78.27 | 73.17 | **97.83** | **95.03** | 40.46 | 47.34 | 64.98 | 73.27 | 70.96 | 78.18 | 63.71 | 67.1 |
| | X-VLM | 50.63 | 65.91 | 53.23 | 65.65 | **95.91** | **93.32** | 65.51 | 74.72 | 75.69 | 81.93 | 57.69 | 67.28 |
| | CLIP$_{ViT}$ | 49.88 | 53.39 | 49.47 | 52.21 | 47.77 | 48.52 | **95.5** | **97.01** | 83.05 | 85.38 | 50.97 | 57.93 |
| | CLIP$_{CNN}$ | 34.78 | 50.42 | 37.17 | 51.24 | 36.81 | 50.87 | 63.07 | 70.92 | **92.48** | **93.66** | 41.81 | 55.12 |
| | BLIP | 54.45 | 55.51 | 55.63 | 53.02 | 41.07 | 46.93 | 61.69 | 69.24 | 65.52 | 75.23 | **83.19** | **82.17** |
| MSCOCO → Flickr30K | ALBEF | **88.08** | **87.28** | 58.9 | 61.53 | 17.58 | 36.07 | 39.78 | 61.08 | 47.28 | 64.95 | 35.02 | 49.4 |
| | TCL | 47.58 | 53.7 | **87.27** | **83.55** | 18.6 | 34.45 | 51.85 | 72.22 | 59.46 | 76.09 | 37.75 | 53.08 |
| | X-VLM | 25.39 | 46.74 | 27.33 | 49.13 | **79.98** | **81.72** | 42.73 | 66.48 | 59.46 | 73.07 | 31.65 | 51.48 |
| | CLIP$_{ViT}$ | 21.07 | 39.47 | 24.53 | 42.44 | 15.45 | 36.52 | **93.97** | **95.53** | 62.95 | 77.21 | 25.55 | 45.91 |
| | CLIP$_{CNN}$ | 14.29 | 36.57 | 20.5 | 41.7 | 15.55 | 37.06 | 41.63 | 62.87 | **86.53** | **88.73** | 17.98 | 44.31 |
| | BLIP | 33.2 | 46.07 | 36.02 | 47.97 | 23.58 | 38.48 | 43.97 | 65.3 | 56.35 | 71.08 | **71.91** | **73.62** |

**Ablation study of the text perturbation.** We introduce another variant C-PGC$_t$ that cancels the perturbation from the text side to investigate the contribution of image perturbation, text perturbation, and their synergy. The comparison results of C-PGC, C-PGC$_t$, and GAP using ALBEF as the surrogate model are shown in Table 13.

Table 13: ASR (%) of C-PGC, C-PGC$_t$, and GAP on ITR tasks using Flickr30K. Note that C-PGC$_t$ only considers attacking images and thus doesn't apply textual perturbations.

| Method | ALBEF | | TCL | | X-VLM | | CLIP$_{ViT}$ | | CLIP$_{CNN}$ | | BLIP | |
|---|---|---|---|---|---|---|---|---|---|---|---|---|
| | TR | IR | TR | IR | TR | IR | TR | IR | TR | IR | TR | IR |
| GAP | 69.78 | 81.59 | 22.15 | 29.97 | 6.61 | 18.37 | 23.4 | 37.54 | 29.92 | 44.29 | 16.09 | 28.12 |
| C-PGC$_t$ | 86.74 | 86.3 | 50.1 | 50.2 | 10.87 | 21.53 | 26.28 | 39.3 | 33.42 | 48.32 | 31.55 | 36.77 |
| C-PGC | **90.13** | **88.82** | **62.11** | **64.48** | **20.53** | **39.38** | **43.1** | **65.93** | **54.4** | **72.51** | **44.79** | **56.36** |

We find that when merely applying image perturbations (C-PGC$_t$), our design still outperforms GAP with notable improvements, validating the proposed techniques' effectiveness in enhancing the image perturbation. Moreover, the superiority of C-PGC over C-PGC$_t$ indicates that the incorporation of textual perturbations can further boost the universal attacks on the basis of C-PGC$_t$ since the text perturbation facilitates the deconstruction of the learned cross-modal alignment.

**Results of R@5 and R@10.** As aforementioned, we supplement the ASR of the ITR tasks based on R@5 and R@10 metrics and provide the attack success rates in Table 14. Obviously, our proposed C-PGC still consistently attains better performance than the baseline method GAP, regardless of the evaluation measurements for retrieval results.

Table 14: Attack success rates (%) regarding R@5 and R@10 metrics of our C-PGC and GAP for image-text retrieval tasks.

| Dataset | Source | Method | ALBEF | | TCL | | X-VLM | | CLIP$_{ViT}$ | | CLIP$_{CNN}$ | | BLIP | |
|---|---|---|---|---|---|---|---|---|---|---|---|---|---|---|
| | | | TR | IR | TR | IR | TR | IR | TR | IR | TR | IR | TR | IR |
| Flickr30K (R@5) | ALBEF | GAP | 55.71 | 73.86 | 8.01 | 10.54 | 1.2 | 4.84 | 4.46 | 15.24 | 8.28 | 20.27 | 5.33 | 10.77 |
| | | Ours | 83.67 | 80.02 | 41.84 | 42.18 | 6.9 | 17.19 | 18.34 | 41.03 | 26.22 | 49.42 | 24.25 | 34.59 |
| | TCL | GAP | 17.64 | 20.09 | 77.89 | 74.53 | 0.9 | 4.2 | 4.25 | 15.48 | 8.6 | 20.25 | 8.65 | 13.16 |
| | | Ours | 29.76 | 35.62 | 90.89 | 84.18 | 3.2 | 13.65 | 20.93 | 42.06 | 25.27 | 49.1 | 16.5 | 30.32 |
| | X-VLM | GAP | 6.21 | 7.45 | 4.9 | 7.96 | 81.6 | 77.23 | 6.11 | 18.33 | 17.41 | 28.35 | 5.03 | 8.61 |
| | | Ours | 7.62 | 25.1 | 8.71 | 26.63 | 89.2 | 85.84 | 19.38 | 42.48 | 30.89 | 50.7 | 13.68 | 29 |
| | CLIP$_{ViT}$ | GAP | 2.81 | 6.86 | 4.2 | 8 | 1.7 | 6.1 | 75.64 | 82.56 | 24.2 | 37.68 | 4.33 | 9.97 |
| | | Ours | 6.31 | 17.51 | 8.01 | 19.65 | 4.3 | 15.1 | 76.89 | 85.2 | 39.6 | 54.68 | 9.15 | 23.23 |
| | CLIP$_{CNN}$ | GAP | 2.81 | 7.41 | 5.81 | 9.27 | 2.4 | 7.01 | 9.33 | 19.64 | 57.63 | 69.33 | 4.02 | 9.76 |
| | | Ours | 3.01 | 19.09 | 5.11 | 22.7 | 3.3 | 23.07 | 17.41 | 41.17 | 61.57 | 74.32 | 6.74 | 25.16 |
| | BLIP | GAP | 3.41 | 7.01 | 3.7 | 7.45 | 1 | 4.4 | 4.46 | 14.43 | 6.79 | 18.67 | 39.13 | 68.02 |
| | | Ours | 14.43 | 21.67 | 13.91 | 21.59 | 5.4 | 14.54 | 18.03 | 36.26 | 23.89 | 44.79 | 59.26 | 74.82 |
| MSCOCO (R@5) | ALBEF | GAP | 74.43 | 78.62 | 37.99 | 30.08 | 5.56 | 7.19 | 14.26 | 17.11 | 15.58 | 21.62 | 23.73 | 23.26 |
| | | Ours | 93.36 | 91.56 | 70.76 | 62.31 | 19.97 | 30.46 | 41.58 | 51.23 | 44.14 | 55.98 | 41.08 | 49.22 |
| | TCL | GAP | 41.48 | 32.59 | 92.54 | 87.81 | 6.46 | 8.08 | 16.09 | 18.47 | 17.98 | 24.3 | 29.9 | 28.64 |
| | | Ours | 60.62 | 56.21 | 94.89 | 90.33 | 22.08 | 30.38 | 53.14 | 64.98 | 58.85 | 70.77 | 45.28 | 53.55 |
| | X-VLM | GAP | 12.29 | 11.64 | 13.43 | 10.99 | 90.8 | 83.05 | 20.02 | 23.4 | 37.72 | 40.09 | 12.64 | 12.04 |
| | | Ours | 31.59 | 48.69 | 32.1 | 48.11 | 96.7 | 91.66 | 49.53 | 60.82 | 59.83 | 69.59 | 37.4 | 52.5 |
| | CLIP$_{ViT}$ | GAP | 18.78 | 17.46 | 20.38 | 17.37 | 16.81 | 15.15 | 95.21 | 93.04 | 62.77 | 62.62 | 18.48 | 19.48 |
| | | Ours | 25.69 | 35.95 | 24.69 | 33.14 | 21.37 | 31.38 | 96.7 | 96.49 | 70.76 | 77.86 | 28.72 | 42.01 |
| | CLIP$_{CNN}$ | GAP | 13.54 | 14.09 | 14.42 | 14.14 | 11.02 | 12.03 | 25.27 | 24.98 | 88.67 | 88.83 | 12.8 | 14.98 |
| | | Ours | 16.25 | 30.07 | 20.96 | 34.15 | 16.58 | 30.23 | 48.04 | 56.66 | 91.54 | 88.96 | 21.37 | 38.85 |
| | BLIP | GAP | 23.62 | 24.22 | 22.96 | 18.43 | 9.93 | 9.75 | 19.2 | 22.24 | 24.99 | 30.19 | 62.75 | 66.88 |
| | | Ours | 42.56 | 43.73 | 41.72 | 41.8 | 31.05 | 35.63 | 44.37 | 57.9 | 54.47 | 66.01 | 81.71 | 81.91 |
| Flickr30K (R@10) | ALBEF | GAP | 51.6 | 71.17 | 5.8 | 6.65 | 0.6 | 2.7 | 1.42 | 9.82 | 4.19 | 13.81 | 3.71 | 7.01 |
| | | Ours | 80.5 | 75.17 | 34.8 | 34.28 | 4.2 | 11.72 | 9.83 | 31.14 | 16.87 | 39.40 | 18.76 | 27.08 |
| | TCL | GAP | 14.9 | 14.29 | 73.26 | 70.49 | 0.6 | 2.22 | 2.13 | 9.73 | 13.45 | 4.29 | 5.92 | 9.51 |
| | | Ours | 24.2 | 27.32 | 89.2 | 80.73 | 2.1 | 9.33 | 12.77 | 32.4 | 16.97 | 38.63 | 12.54 | 24.1 |
| | X-VLM | GAP | 4.1 | 4.81 | 2.7 | 4.51 | 76.5 | 72.58 | 3.65 | 11.63 | 10.84 | 18.91 | 3.01 | 5.29 |
| | | Ours | 4.1 | 17.79 | 4.6 | 19.27 | 86.3 | 82.94 | 11.14 | 31.49 | 21.06 | 40.3 | 7.32 | 21.81 |
| | CLIP$_{ViT}$ | GAP | 2.1 | 4.04 | 2.6 | 4.81 | 1 | 3.79 | 63.29 | 77.83 | 17.89 | 28.11 | 2.31 | 6.12 |
| | | Ours | 4.2 | 11.45 | 4.6 | 13.08 | 2.8 | 10.15 | 67.98 | 79.46 | 29.75 | 45.56 | 5.52 | 17.23 |
| | CLIP$_{CNN}$ | GAP | 2.3 | 4.23 | 3.3 | 5.56 | 1.4 | 4.41 | 4.56 | 12.35 | 49.86 | 62.17 | 2.21 | 6.63 |
| | | Ours | 2.5 | 13.56 | 3.7 | 16.87 | 1.7 | 17.38 | 9.93 | 32.4 | 53.27 | 66.34 | 3.91 | 19.62 |
| | BLIP | GAP | 2 | 3.98 | 1.5 | 4.17 | 0.2 | 2.24 | 1.93 | 8.38 | 3.68 | 12.48 | 36.81 | 67.22 |
| | | Ours | 11.2 | 15.49 | 9.1 | 14.14 | 2.8 | 10.19 | 9.83 | 27.46 | 14.83 | 34.43 | 53.46 | 72 |
| MSCOCO (R@10) | ALBEF | GAP | 69.78 | 76.24 | 32.04 | 23.7 | 3.16 | 4.91 | 10.07 | 13.16 | 12.55 | 16.6 | 19.1 | 19.96 |
| | | Ours | 91.58 | 89.62 | 64.5 | 55.3 | 13.81 | 23.34 | 33.3 | 43.75 | 35.82 | 48.38 | 33.77 | 43.11 |
| | TCL | GAP | 34.36 | 25.76 | 90.65 | 85.27 | 4.01 | 5.44 | 11.77 | 14.88 | 13.66 | 18.89 | 24.91 | 24.67 |
| | | Ours | 52.59 | 49.09 | 93.63 | 88.53 | 15.04 | 23.25 | 44.22 | 58.02 | 50.16 | 63.95 | 37.77 | 47.26 |
| | X-VLM | GAP | 7.66 | 7.65 | 8.07 | 7.27 | 88.3 | 79.85 | 15.63 | 18.77 | 31.79 | 33.54 | 8.24 | 9.37 |
| | | Ours | 23.01 | 40.39 | 23.15 | 40.07 | 94.97 | 88.95 | 40.24 | 53.74 | 52 | 62.7 | 30.43 | 45.67 |
| | CLIP$_{ViT}$ | GAP | 13.13 | 12.39 | 14.05 | 12.13 | 10.68 | 10.89 | 93.72 | 91.51 | 57.39 | 56.47 | 13.35 | 15.64 |
| | | Ours | 17.87 | 28.52 | 17.48 | 26.09 | 14 | 24.67 | 95.55 | 95.31 | 64.04 | 72.75 | 22.05 | 35.65 |
| | CLIP$_{CNN}$ | GAP | 9.02 | 10.08 | 9.06 | 9.89 | 6.97 | 8.25 | 18.78 | 20.11 | 87.6 | 83.92 | 8.53 | 11.91 |
| | | Ours | 10.68 | 22.98 | 14.19 | 27.21 | 10.62 | 23.96 | 39.89 | 49.62 | 88.74 | 85.18 | 15.97 | 33.25 |
| | BLIP | GAP | 17.76 | 18.98 | 16.32 | 13.13 | 6.21 | 6.44 | 13.97 | 17.39 | 19.73 | 24.37 | 57.99 | 65.49 |
| | | Ours | 33.64 | 36.14 | 32.15 | 33.8 | 22.64 | 28.52 | 36.07 | 50.3 | 47 | 59.07 | 78.39 | 78.98 |

# F    MULTIMODAL ALIGNMENT DESTRUCTION

To provide more intuitive evidence that our C-PGC successfully destroys the image-text alignment relationship, we compute the distance between the encoded image and text embeddings before and

after applying the UAP. Concretely, for an input pair (v, t), we calculate the relative distance $d_{rel}$ by:

$$d_{rel} = \frac{||(f_I(v + \delta_v) - f_T(t \oplus \delta_t)||_2 - ||f_I(v) - f_T(t)||_2}{||f_I(v) - f_T(t)||_2}. \quad (6)$$

We provide the distances averaged on 5000 image-text pairs from Flickr30K in Table 15. Benefiting from our delicate designs, C-PGC achieves better disruption of the aligned multimodal relationship, thereby boosting the generalization ability and transferability of the produced UAP.

Table 15: Relative cross-modal feature distances to the clean image-text pairs.

| Source | Method | ALBEF | TCL | BLIP | X-VLM | CLIP$_{ViT}$ | CLIP$_{CNN}$ |
|--------|--------|-------|-----|------|-------|------|------|
| ALBEF | GAP | 7.18 | 6.54 | 0.91 | 1.74 | 0.31 | 0.98 |
|  | C-PGC | **8.83** | **14.95** | **2.73** | **6.09** | **3.42** | **3.92** |
| TCL | GAP | 4.02 | 24.27 | 0.91 | 0.87 | 0.12 | 0.07 |
|  | C-PGC | **6.43** | **27.11** | **3.64** | **4.35** | **2.56** | **2.94** |
| BLIP | GAP | 3.17 | 4.67 | 11.82 | 1.74 | -1.71 | -0.98 |
|  | C-PGC | **6.41** | **12.15** | **13.64** | **4.35** | **1.71** | **1.96** |

## G  MORE TRAINING DETAILS

For Flickr30K and MSCOCO, we randomly sample 30,000 images and their captions from the training set to train our perturbation generator. For SNLI-VE and RefCOCO+, we learn the C-PGC directly using their training set with 29,783 and 16,992 images respectively. Since an image corresponds to multiple text descriptions in these datasets, we calculate the average of their textual embedding as the multimodal condition for the cross-attention modules.

We initialize the noise variable $z_v$ as a $3 \times 3$ matrix. Meanwhile, the initial noise $z_t$'s dimensions in the text modality depend on the size of the hidden layer within the specific VLP model. Concretely, we set its dimension to $1 \times 3$ for ALBEF, TCL, BLIP, and X-VLM, while $1 \times 2$ for the CLIP model. When computing the multimodal contrastive loss $\mathcal{L}_{CL}$, the temperature $\tau$ is set as 0.1. The generator is trained over 40 epochs with the Adam optimizer, utilizing a learning rate of $2^{-4}$. Following previous works Lu et al. (2023); Wang et al. (2024), we employ the attack success rate (ASR) as our quantitative measurement of our attack in ITR tasks by computing the extent the adversarial perturbations result in victim models' performance deviations from the clean performance.

## H  DETAILED INTRODUCTIONS TO DATASETS

- **Flickr30K (Plummer et al., 2015).** Collected from the Flickr website, this dataset describes different items and activities, which becomes a standard benchmark for various V+L tasks. It contains 31,783 images, each of which has five associated captions. We use it for ITR tasks.

- **MSCOCO (Lin et al., 2014).** The MSCOCO dataset is a rich and diverse dataset consisting of 123,287 images, each of which is annotated with approximately five sentences. We use this dataset to test the attack performance of ITR and IC tasks.

- **SNLI-VE (Xie et al., 2019).** Originally proposed for natural language reasoning tasks, this dataset provides large-scale images and descriptions, where each image is annotated with several sentences and their logical relationship labels, including entailment, neutral, and contradiction. This dataset is used for VE tasks.

- **RefCOCO+ (Yu et al., 2016).** An image dataset was selected from MSCOCO, which contains 19,992 images and 141,564 annotations. It is specially used for visual grounding (VG) tasks.

## I  DISCUSSIONS AND FUTURE DIRECTIONS

**Overlook of interactions between perturbations $\delta_v$ and $\delta_t$.** The proposed framework generates universal perturbations for image and text respectively based on the designed multimodal and uni-modal losses. Despite the remarkable attack performance, it does not consider the interactions and

synergy between the perturbations $\delta_v$ and $\delta_t$ during optimization, which has been leveraged in several previous attacks (Lu et al., 2023; Wang et al., 2024) to improve performance. In future research, this limitation can be explored as a potential mechanism to further strengthen the attacks.

**Textual Semantic consistency.** To ensure the stealthiness of text attacks, we set the perturbation budget $\epsilon_t = 1$, i.e., only one word is modified. Despite the superior semantic similarity to previous sample-specific methods, there is still room to improve from the proposed C-PGC. Moreover, future studies can consider similarity preservation strategies by applying more effective constraints during the generator training or post-processing adversarial sentences to facilitate a more stealthy attack.

**Leveraging Task-level Characteristics.** In contrast to the unimodal scenarios, we fully leverage the unique characteristics of multimodal scenarios to enhance the modeling of a universal perturbation that can effectively generalize to diverse downstream V+L tasks. While this work lies in leveraging the shared and joint characteristics of Vision-Language scenarios to present a universal and versatile UAP, it is a promising direction for future studies to investigate task-level V+L characteristics to further enhance attacks for specific downstream tasks.

**Synthetic positive samples.** Introducing synthetic samples that are maximally distant from the anchor as positive samples is a promising direction. A reasonable implementation might involve adversarial learning to generate such maximally distant samples. However, this strategy necessitates synthesizing samples for each input pair, leading to a significant increase in the computational overhead. Future works can explore more efficient and effective positive sample strategies.

## J   SPECIAL TOKENS AS TEXT PERTURBATION

We also explore the potential of special tokens to serve as adversarial perturbations. Specifically, we directly adopt two typical # and * as adversarial tokens to evaluate their attack results.

Table 16: ASR of C-PGC and its variants using special characters as the adversarial word.

| Source | Adv. word | ALBEF | | TCL | | X-VLM | | CLIP$_{ViT}$ | | CLIP$_{CNN}$ | | BLIP | |
|---|---|---|---|---|---|---|---|---|---|---|---|---|---|
| | | TR | IR | TR | IR | TR | IR | TR | IR | TR | IR | TR | IR |
| ALBEF | # | 87.81 | 85.74 | 60.84 | 62.05 | 18.28 | 35.87 | 38.67 | 61.4 | 50.91 | 68.21 | 41.71 | 54.11 |
| | * | 87.21 | 84.87 | 60.24 | 62.19 | 18.01 | 35.4 | 38.79 | 61.91 | 51.27 | 68.1 | 41.92 | 54.69 |
| | C-PGC | **90.13** | **88.82** | **62.11** | **64.48** | **20.53** | **39.38** | **43.1** | **65.93** | **54.4** | **72.51** | **44.79** | **56.36** |
| CLIP$_{ViT}$ | # | 21.25 | 37.41 | 24.27 | 41.04 | 14.71 | 34.38 | 87.07 | 92.39 | 63.2 | 75.14 | 25.46 | 44.19 |
| | * | 22.07 | 37.54 | 24.58 | 41.32 | 14.81 | 34.77 | 87.57 | 92.41 | 63.78 | 74.86 | 25.76 | 44.87 |
| | C-PGC | **23.23** | **38.67** | **25.05** | **41.79** | **15.85** | **35.59** | **88.92** | **93.05** | **66.06** | **75.42** | **26.71** | **45.7** |

Table 17: Comparison of C-PGC and its variants using special characters as the adversarial word regarding the BERT score between clean and adversarial texts.

| Adv. word | ALBEF | | | TCL | | | CLIP_VIT | | | CLIP_CNN | | |
|---|---|---|---|---|---|---|---|---|---|---|---|---|
| | P | R | F1 | P | R | F1 | P | R | F1 | P | R | F1 |
| # | 0.8213 | 0.8419 | 0.8313 | 0.8171 | 0.8389 | 0.8277 | 0.8137 | 0.8339 | 0.8235 | 0.8156 | 0.8364 | 0.8257 |
| * | 0.8149 | 0.8251 | 0.8197 | 0.8098 | 0.8206 | 0.8149 | 0.8095 | 0.8206 | 0.8148 | 0.8097 | 0.8203 | 0.8147 |
| C-PGC | **0.8891** | **0.8613** | **0.8748** | **0.8924** | **0.8687** | **0.8802** | **0.8746** | **0.8684** | **0.8713** | **0.8948** | **0.8842** | **0.8893** |

Table 16 and Table 17 display that our optimization-based strategy exhibits both superior attack performance and higher semantic similarity. Future studies can investigate more special tokens to increase the likelihood of bypassing human observers or automated filtering systems.

## K   MORE VISUALIZATION RESULTS

This section presents a rich visual analysis of the proposed attack on a series of downstream tasks. Specifically, we generate the UAP and conduct attacks on the ALBEF model in the visual grounding (VG) task. As illustrated in Figure 8, the prediction bounding boxes exhibit a notable deviation from the clean predictions, verifying that our generated adversarial samples significantly interfere with the multimodal alignment. In the visual entailment (VE) task, we employ BLIP as the victim model and present the results in Figure 9. These qualitative visualizations again demonstrate the remarkable attack effects of our proposed method on various downstream tasks.

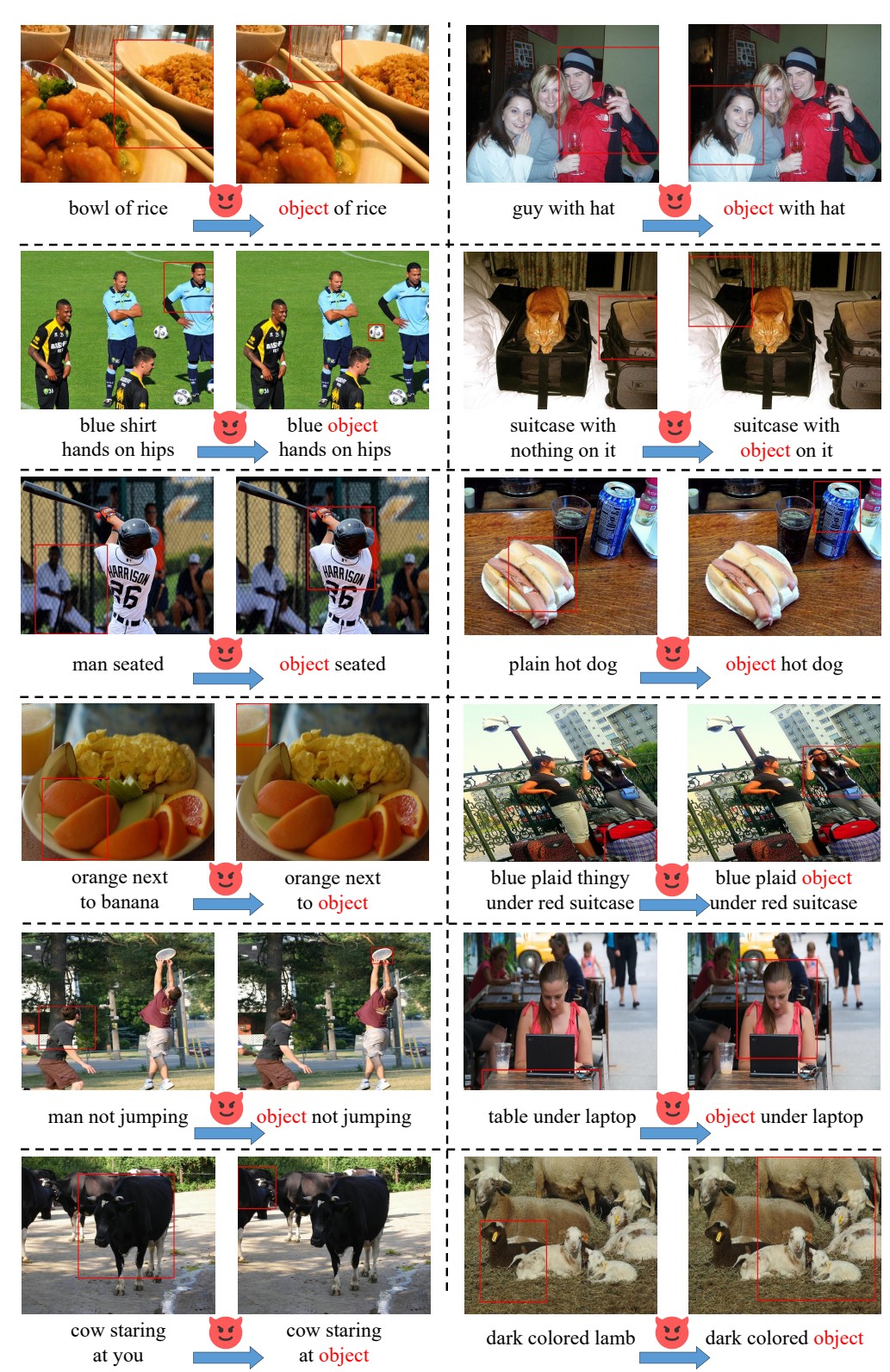

Figure 8: Illustration of visual grounding. Predictions of clean image-text pairs are on the left while the adversarial samples are on the right. The red indicates the universal adversarial word.

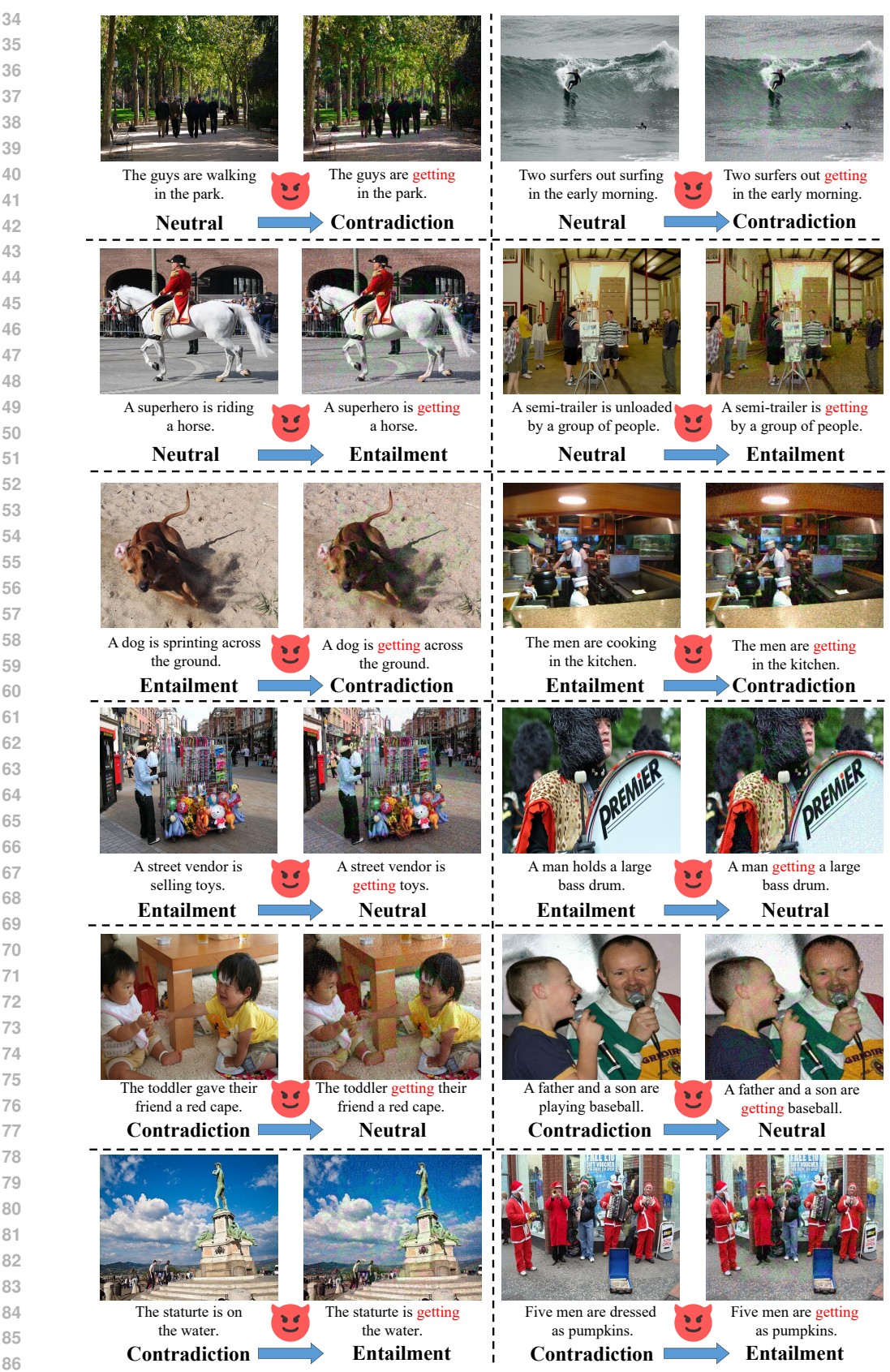

Figure 9: Illustration of the visual entailment task. The red indicates the universal adversarial word. It can be observed that all predictions do not match with the ground truth.

