# OpenReview forum: "One Perturbation is Enough: On Generating Universal Adversarial Perturbations against Vision-Language Pre-training Models"
_ICLR.cc/2025/Conference — Submitted to ICLR 2025_

### Official Review · Reviewer_4NRc · 2024-10-19

**Soundness:** 4
**Presentation:** 4
**Contribution:** 3
**Rating:** 6
**Confidence:** 5

**Summary:**

This paper introduces a universal adversarial attack method called C-PGC for VLP models. By leveraging contrastive learning mechanisms within VLP models, C-PGC generates UAP that can effectively attack across different tasks and models without requiring individual perturbations for each input. Experiments demonstrate that C-PGC disrupts image-text feature alignment efficiently in both white-box and black-box settings, outperforming existing attack methods.

**Strengths:**

1.	The writing is clear. The formulas are correct.
2.	The experiment is abundant and multi-dimensional.
3.	The research topic is important for VLM.

**Weaknesses:**

1. The generator-based UAP method is time-consuming due to its indirect optimization approach, as it does not directly update the UAP.
2. In multimodal contrastive loss, randomly selecting texts or images may not be ideal. Instead, selecting items related to the current image-text pair in the batch could improve the performance.
3. The method uses the default settings of SGA in the experiment (i.e., resizing the original images into five scales {0.5, 0.75, 1, 1.25, 1.5} and applying Gaussian noise with a mean of 0 and a standard deviation of 0.5). It would be beneficial to give an explanation of the effect of such augmentation, with the experiment result being better.

**Questions:**

Please see weakness

**Details Of Ethics Concerns:**

The paper is related to adversarial attack, which is an important subarea in AI security

---

> ### Author Response · Authors · 2024-11-20
> **Author Response**
>
> Thank you for dedicating your time and effort to reviewing our paper. We are deeply encouraged by your positive comments and recognition of our work. Below, we provide point-by-point responses to address your concerns.
>
> > `[Q1]` The time budget of generator training.
>
> Yes! Training a generator can be time-consuming and is a common issue in the whole research field of generative attacks [1,2,3]. However, the generative paradigm can bring significant performance improvement and once the generator finishes training, the universal perturbation can be **directly applied to any image-text pair without requiring any sample-specific optimization**.
>
> > `[Q2]` Positive sample selection.
>
> We are deeply sorry for the misunderstandings that our paper may cause you. Please kindly allow us to classify the positive sample selection more clearly.
> Taking image perturbation as an example, we first randomly sample a batch of texts as candidates, where we then **choose texts with the farthest distance specific to the current input adversarial image** as positive samples.
> *i.e.*, our framework has taken the current input pair into consideration when selecting items for contrastive loss. The ablation study in **Section 4.4** of the main body has verified the effectiveness of this farthest selection strategy.
>
> We will highlight this design to make it more clear. Also, we will include your constructive suggestion as a potential direction for a better selection strategy in the *Discussion* section of the Appendix.
>
> > `[Q3]` The effect of set-level augmentation.
>
> Yes. As described in the method part, we are motivated by the significant gains introduced by SGA's augmentation technique [4] and hence integrate it into the proposed framework to enhance the universal perturbation. The underlying mechanism is to leverage the many-to-many relationships between images and texts by introducing multiple augmented images to provide set-level diverse guidance, further improving the optimization direction for effective guidance.
> Following your suggestion, we conduct experiments where this augmentation strategy was removed from our method.
>
> Table A. Comparison of C-PGC and its variant that cancels the data augmentation on Flickr30.
> | Source   | Target$\Rightarrow$ | ALBEF |    ALBEF   | TCL   | TCL      | X-VLM  |  X-VLM     | CLIP_VIT |   CLIP_VIT    | CLIP_CNN |  CLIP_CNN     | BLIP  |  BLIP     |
> |----------|--------|-------|-------|-------|-------|-------|-------|----------|-------|----------|-------|-------|-------|
> |    $\Downarrow$      |   Method $\Downarrow$    | TR    | IR    | TR    | IR    | TR    | IR    | TR       | IR    | TR       | IR    | TR    | IR    |
> | ALBEF    | w/o Aug | 69.78 | 74.79 | 47    | 57.26 | 20.43 | 37.55 | 42.36    | 65.17 | 53.63    | 71.6  | 41.22 | 55.34 |
> |    ALBEF      | Ours   | **90.13** | **88.82** | **62.11** | **64.48** | **20.53**| **39.38** | **43.1**     | **65.93** | **54.4**     | **72.51** | **44.79** | **56.36** |
> | CLIP_VIT | w/o Aug | 18.5  | 37.8  | 22.19 | 39.86 | 13.47 | 34.17 | 86.46    | 87.11 | 61.53    | 71.36 | 25.03 | 44.73 |
> |    CLIP_VIT      | Ours   | **23.23** | **38.67** | **25.05** | **41.79** | **15.85** | **35.59** | **88.92**    | **93.05** | **66.06**    | **75.42** | **26.71**| **45.7**  |
>
> Table A demonstrates that the augmentation mechanism indeed provides an improvement in ASR, proving the rationality of set-level augmentation. We will include these results in **Appendix E** to complement our ablation studies.
>
> Thank you again for your thoughtful suggestion! If you have any further concerns, feel free to reach out to us. :)
>
> [1] Feng W, Xu N, Zhang T, et al. Dynamic generative targeted attacks with pattern injection. In CVPR, 2023.
>
> [2] Poursaeed O, Katsman I, Gao B, et al. Generative adversarial perturbations. In CVPR, 2018.
>
> [3] Gao H, Zhang H, Wang J, et al. NUAT-GAN: Generating Black-box Natural Universal Adversarial Triggers for Text Classifiers Using Generative Adversarial Networks. In IEEE TIFS, 2024.
>
> [4] Lu D, Wang Z, Wang T, et al. Set-level guidance attack: Boosting adversarial transferability of vision-language pre-training models. In ICCV, 2023.

---

### Official Review · Reviewer_U4zG · 2024-10-21

**Soundness:** 3
**Presentation:** 2
**Contribution:** 2
**Rating:** 5
**Confidence:** 5

**Summary:**

The paper presents a novel framework, C-PGC, designed to generate Universal Adversarial Perturbations (UAPs) targeting Vision-Language Pretraining (VLP) models. The authors introduce a cross-modal conditional perturbation generator, which leverages both single-modal and cross-modal features to disrupt the learned alignment between visual and textual representations in VLP models.

**Strengths:**

The writing style of the paper is commendably clear and concise, making it accessible to a broad audience within the machine learning and computer vision communities. The authors have taken care to present the technical details in a manner that is straightforward and easy to follow, even for readers who may not be deeply familiar with adversarial attacks or VLMs. The method’s components are explained in a way that balances technical rigor with simplicity. This makes the paper highly readable and ensures that a wide range of researchers and practitioners can engage with its contributions.

The experimental results demonstrate that the proposed C-PGC framework performs well across several benchmarks. The authors have conducted comprehensive experiments on multiple well-established datasets and across various VLP models. The results show consistent improvements in attack success rates (ASR) across both white-box and black-box settings.

**Weaknesses:**

The paper, while strong overall, has several areas for improvement:

1. **Use of Contrastive Loss**
   The inclusion of contrastive loss (\(\mathcal{L}_{CL}\)) feels somewhat forced. Since the goal is to perform untargeted attacks, it seems unnecessary to rely on contrastive loss, which is typically used to enforce alignment between representations. While the authors have shown its utility through ablation studies, the logical foundation of using contrastive loss in an untargeted setting remains unclear. The paper could be improved by either rethinking the rationale behind using \(\mathcal{L}_{CL}\) or exploring alternative loss functions better suited for untargeted attacks.

2. **Choice of Positive and Negative Samples for Contrastive Loss**
   The current method of manually selecting the farthest sample as the positive or negative example feels arbitrary and unnecessarily complex. In the context of untargeted attacks, where the objective is not to make the adversarial sample resemble a specific class, it would make more sense to introduce a synthetic, "fictitious" sample that maximally deviates from the original, rather than relying on the farthest feature-distance sample. This approach could simplify the process and make the use of contrastive loss more coherent in the untargeted setting.

3. **Limited Comparison with State-of-the-Art**
   The comparison of the proposed method only with GAP, a 2018 work, limits the scope of the evaluation. Given the rapid advancements in adversarial attack methods, comparing against more recent techniques would provide a clearer picture of the method’s competitiveness. For example, comparing with more contemporary adversarial generation methods would strengthen the experimental section and make the results more relevant to current research.

4. **Visual Design of the Framework Diagram**
   The framework diagram could benefit from improved design and color harmony. While this is a minor issue, it affects the overall presentation quality. The authors could refer to well-designed diagrams from recent top-tier papers to make the visualizations clearer and more appealing.

These adjustments would enhance the logical foundation of the method and improve both the clarity and relevance of the experimental comparisons.

Post-Rebuttal:

The main issue lies in the fact that the authors' response fails to address my concerns regarding the use of Contrastive Loss in the paper. The goal of the paper is to construct untargeted adversarial attacks, yet the authors manually select negative samples to construct the Contrastive Loss, which is problematic. If we follow the authors' stated motivation that "destruction is easier than construction," the approach should involve maximizing the Contrastive Loss during normal training, rather than manually selecting negative samples to minimize it. For these reasons, I am inclined to maintain my current rating.

**Questions:**

Please refer to the **Weaknesses** section. If the authors can address these issues, I would be willing to raise my score.

1. **Use of Contrastive Loss**
Perhaps the authors could offer a more detailed explanation of their rationale for using contrastive loss in this context and discuss potential alternative loss functions they considered, as well as why contrastive loss was ultimately chosen despite its typical use in alignment tasks.

2.  **Choice of Positive and Negative Samples for Contrastive Loss**
Introducing a synthetic, "fictitious" sample that maximally deviates from the original, is a more direct way. Perhaps the authors could discuss the trade-offs between their current approach and your suggested method.

3. **Limited Comparison with State-of-the-Art**
The authors could select 2-3 recent and relevant adversarial attack methods from related work for comparison.

---

> ### Author Response · Authors · 2024-11-20
> **Author Response (Part I)**
>
> We sincerely express our gratitude for dedicating your valuable time to providing insightful suggestions that can enhance our paper. Your praise regarding our writing, methodology, experiments, and contributions has greatly encouraged and motivated us. Our detailed responses to all of your concerns are presented below.
> > `[Q1]` Use of Contrastive Loss.
>
> We sincerely apologize for not adequately explaining the rationale behind employing the contrastive learning mechanism in our paper. We then provide a detailed explanation of the underlying principles of using contrastive loss, supported by more experimental results.
>
> 1. **Motivation**.  It is widely acknowledged that contrastive learning serves as a powerful and foundational tool for modality alignment in VLP models, establishing a nearly point-to-point relationship between image and text features. Our core idea stems from the general principle: *"It's easier to tear down than to build up."* Since contrastive learning can effectively establish robust and precise alignment, leveraging the same technique to disrupt these established alignments is expected to yield effective attack performance.
>
> 2. **Rationale**. Taking image attack as an example, the principle behind our contrastive learning-based attack can be understood from two perspectives. (1) Leverage the originally matched texts as negative samples to push the aligned image-text pair apart. **This broadly corresponds to the common objective of untargeted attacks that you have kindly mentioned.** (2) Additionally, our contrastive paradigm introduces additional benefits by using dissimilar texts as positive samples to pull the adversarial image $v^{adv}$ out of its original subspace and relocate it to an incorrect feature area.
>    By **simultaneously harnessing the collaborative effects of `push` (negative samples) and `pull` (positive samples)**, the proposed contrastive framework effectively destroys the modal alignment and achieves exceptional attack performance, which has been validated by comprehensive experimental results.
>
> Besides, we also explore several potential alternative loss functions that more directly align with the common untargeted attack, including maximizing the negative cosine similarity $\mathcal{L}\_{Cos}$ or MSE distance $\mathcal{L}\_{MSE}$ between the features of matched image-text pairs.
>
> Table A. ASR results of different loss functions when the surrogate is ALBEF.
> |Target$\Rightarrow$ | ALBEF |  ALBEF     | TCL   |   TCL    | X-VLM |  X-VLM     | CLIP_VIT |   CLIP_VIT    | CLIP_CNN |   CLIP_CNN    | BLIP  |   BLIP    |
> |----------|-------|-------|-------|-------|-------|-------|----------|-------|----------|-------|-------|-------|
> |   Method$\Downarrow$       | TR    | IR    | TR    | IR    | TR    | IR    | TR       | IR    | TR       | IR    | TR    | IR    |
> | $\mathcal{L}\_{MSE}$      | 12.02 | 30.75 | 14.39 | 35.08 | 11.41 | 30.79 | 37.32    | 56.05 | 40.17    | 56.39 | 19.66 | 37.33 |
> | $\mathcal{L}\_{Cos}$      | 57.55 | 67.4  | 37.06 | 49.45 | 10.7  | 28.48 | 37.49    | 58.3  | 40.87    | 58.39 | 23.33 | 39.44 |
> | $\mathcal{L}\_{CL}$       | **76.46** | **82.46** | **56.52** | **62.61** | **14.33** | **33.61** | **42.98**    | **62.81** | **46.11**    | **65.58** | **27.13** | **46.44** |
> | $\mathcal{L}\_{MSE}+\mathcal{L}\_{Dis}$  | 81.09 | 83.71 | 48.76 | 56.54 | 17.58 | 35.72 | 41.5     | 64.72 | 47.41    | 70.34 | 35.96 | 51.76 |
> | $\mathcal{L}\_{Cos}+\mathcal{L}\_{Dis}$  | 65.2  | 72.71 | 36.13 | 50.06 | 18.63 | 36.74 | 42.23    | 65.17 | 50.91    | 69.78 | 36.91 | 50.69 |
> | $\mathcal{L}\_{CL}+ \mathcal{L}\_{Dis}$    | **90.13** | **88.82** | **62.11** | **64.48** | **20.53** | **39.38** | **43.1**     | **65.93** | **54.4**     | **72.51** | **44.79** | **56.36** |
>
> Recall that $\mathcal{L}\_{CL}$ and $\mathcal{L}\_{Dis}$ denote the proposed contrastive loss and the unimodal loss term respectively. As observed, the use of $\mathcal{L}\_{CL}$ consistently brings significant ASR improvements, verifying the rationality and superiority of contrastive loss.
>
> We will include these analyses in a new section of the Appendix to supplement a more thorough understanding of the rationale behind the choice of contrastive loss.
> Thank you once again for your inspiring suggestion!

---

> > ### Author Response · Authors · 2024-11-22
> > **Author Response (Part II)**
> >
> > > `[Q2]` Choice of Samples for Contrastive Loss.
> >
> > Very interesting suggestion!  Introducing synthetic samples that are maximally distant from the anchor as positive samples is a promising strategy to enhance attack performance. A reasonable implementation might involve adversarial learning to generate such maximally distant samples. However, this strategy necessitates synthesizing these maximally distant samples for each input pair during the generator training, which can lead to a significant increase in the computational overhead. In contrast, the current sampling and selection method achieves impressive attack performance without imposing significant additional burdens, striking a balance between efficiency and effectiveness.
> >
> > Also, we will include this discussion in the Appendix to encourage future exploration of more efficient and effective positive sample selection strategies.
> >
> > > `[Q3]` Comparison with the State-of-the-Art.
> >
> > Thanks for this thoughtful advice! We supplement a recent study of UAP on VLP models for comparison. Please refer to the `[Q2]` in the **Common Concerns**.
> >
> > > `[Q4]` Visual Design of the Framework Diagram.
> >
> > Thank you for your valuable suggestion! Following your advice, we adjusted the color schemes and visual materials of the framework diagram in the revision of our paper, which will be uploaded soon. Furthermore, we will learn from more top-tier papers to further enhance the paper's expressiveness and appeal.
> >
> > We hope these responses can address your concerns. Once again, we deeply appreciate your valuable suggestions for improving our work and would be delighted to further discuss with you.

---

> > ### Comment · Reviewer_U4zG · 2024-11-25
> > **Response to authors**
> >
> > Thanks for your addressing.
> > I have gotten your point. Have you tried using the negative direction of the contrastive loss, instead of manually selecting negative samples to construct the contrastive loss?

---

> ### Author Response · Authors · 2024-11-23
> **A Kind Reminder of the Author Response**
>
> Dear Reviewer U4zG,
>
> Thank you once again for dedicating your valuable time to reviewing our paper and providing constructive comments!
>
> As the end of the discussion period approaches, we kindly ask if our responses have satisfactorily addressed your concerns. Your feedback would be greatly appreciated, and we would be delighted to engage in further discussions with you.
>
> Sincerely,
>
> The Authors

---

> ### Author Response · Authors · 2024-11-25
> **Author Response (Part III)**
>
> Thanks for your inspiring acknowledgment of our response!
>
> We apologize for not fully understanding the *"negative direction of the contrastive loss, instead of manually selecting negative samples to construct the contrastive loss?"* meant, since it is generally necessary to obtain both positive and negative samples for contrastive loss formulation. Does that indicate our proposed strategy in the response in `[Q2]`? The experiments are close to end and we will upload the results very soon.
>
> We would appreciate it if you could provide more details so that we can more accurately address this matter. :)

---

> > ### Comment · Reviewer_U4zG · 2024-11-26
> > **response to author**
> >
> > maximum the contrastive loss

---

> ### Author Response · Authors · 2024-11-26
> **Looking forward to more discussions**
>
> Dear U4zG,
>
> We have carefully supplemented responses to your further questions and provided experiments following your suggestion.
> We look forward to your reply and welcome discussion on any questions regarding our paper and response.
>
> Best regards,
>
> Authors

---

> ### Author Response · Authors · 2024-11-26
> **Author Response (Part V)**
>
> We are very sorry that we have not made the attempt to maximize the contrastive loss, since maximizing the contrastive loss seems to be contrary to our attack goal.
>
> Could you please tell us more details about the purpose of this experiment?  Also, we wonder if the supplemented experiments to `[Q2]` in **Author Response (Part IV)** have adequately solved your issue.
>
> Thanks for your response!

---

> ### Author Response · Authors · 2024-11-27
> **Author Response (Part VI)**
>
> Dear U4zG,
>
> We apologize for still not fully understanding the aim of *"maximum the contrastive loss"*. Therefore, we clarify the following facts as potential answers to your raised question. Please kindly review them to see if this response has solved your concern.
>
> 1. We highlight that **it requires both positive and negative samples to formulate the contrastive learning paradigm**. Generally, positive samples are defined as the targets that the anchor aims to move closer to, while negative samples are those that it seeks to move away from [1,2,3].
> 2. Maximizing the established contrastive loss **violates our attack goal**, since this operation actually **pushes the adversarial image closer to the original matched texts (negative samples) and pulls the adversarial image away from the dissimilar texts (positive samples)**. This also fails to obey the definition of contrastive learning, since the anchor sample is supposed to get away from the negative samples and closer to the positive samples.
> 3. We provide results where we directly maximize the cosine similarity and MSE between the adversarial sample and its paired data in **Table A of Author Response (Part I)**. Also, we have provided experiments using synthetic samples based on adversarial learning in **Table B of Author Response (Part IV)**.
> 4. Do you actually indicate maximizing the distance between the anchor sample and the positive samples?
>
> We sincerely hope that you can tell us **whether the above responses have adequately solved your concerns** or provide more details about your question. We're glad to have further discussion with you. If you are satisfied with our responses, we would greatly appreciate it if you could kindly consider raising your score accordingly. :)
>
> We're looking forward to your reply.
>
> Sincerely,
>
> Authors
>
> [1] Khosla P, Teterwak P, Wang C, et al. Supervised contrastive learning. In NIPS, 2020.
>
> [2] Li J, Selvaraju R, Gotmare A, et al. Align before fuse: Vision and language representation learning with momentum distillation. In NIPS, 2021.
>
> [3] Yang J, Duan J, Tran S, et al. Vision-language pre-training with triple contrastive learning. In CVPR, 2022.

---

> > ### Author Response · Authors · 2024-12-02
> > **A Gentle Reminder of the End of Rebuttal**
> >
> > Dear Reviewer U4zG,
> >
> > This is a gentle reminder that the rebuttal period is approaching its conclusion. Since we have addressed your major concerns **(Q1, Q2, Q3, Q4)** during the rebuttal, we would like to kindly ask if you could consider raising your score. :)
> >
> >
> > Best regards,
> >
> > Authors

---

> ### Author Response · Authors · 2024-12-03
> **The Last Gentle Reminder of the End of Rebuttal**
>
> Dear Reviewer U4zG,
>
> Since the discussion phase will last **nearly only one hour**, we kindly send this message as the last gentle reminder of our responses.
>
> As we have addressed your major concerns **(Q1, Q2, Q3, Q4)** during the rebuttal, we would like to kindly ask if you could consider raising your score. :)
>
>
> Best regards,
>
> Authors

---

### Official Review · Reviewer_AoYQ · 2024-11-01

**Soundness:** 3
**Presentation:** 3
**Contribution:** 3
**Rating:** 6
**Confidence:** 4

**Summary:**

This paper addresses the vulnerability of Vision-Language Pre-training (VLP) models to universal adversarial perturbations, which are instance-agnostic and do not require individual perturbations for each input. The authors introduce a novel attack method, the Contrastive-training Perturbation Generator with Cross-modal conditions (C-PGC), which leverages contrastive learning to disrupt the multimodal alignment in VLP models. Experiments are conducted across multiple VLP models and tasks.

**Strengths:**

1. The proposed UAP framework addresses the inefficiencies of instance-specific attacks by incorporating cross-modal and unimodal guidance within a contrastive training setup, representing an advancement in universal adversarial attack methods.

2. The paper thoroughly evaluates C-PGC's effectiveness across multiple VLP models and downstream tasks, and additionally analyzes various defense strategies to mitigate the potential risks posed by C-PGC.

**Weaknesses:**

The proposed method leverages image and text attacks alongside cross-modal contrastive learning to generate universal adversarial perturbations. While this approach shows promise, the novelty may not be fully evident. I recommend that the authors consider further highlighting and reorganizing the unique contributions of the paper to enhance its clarity and impact.

**Questions:**

1. In the text modality attack, how do the authors maintain semantic similarity between original and adversarial texts? In the experiments, the authors should provide the similarity scores (e.g. bert_score) between original and adversarial texts to demonstrate that the modifications do not significantly alter the text's semantics.

2. The authors should explain how ASR is calculated in the main text.

3. As shown in Figure 4, the adversarial texts exhibit a clear semantic gap from the original texts. Thus, would using special characters (e.g. ##*) for the universal adversarial word be more effective?


4. The authors propose a contrastive training perturbation generator to produce universal adversarial perturbations for images and text. I am curious about how this generator differs from general UAP methods (e.g. Data-free Universal Adversarial Perturbation and Black-box Attack ), justifying its designation as a "generator."

I look forward to your detailed response.

---

> ### Author Response · Authors · 2024-11-20
> **Author Response (Part I)**
>
> We would like to express our gratitude for your valuable suggestions and positive feedback on our paper. Our detailed responses are provided below.
>
> > `[Q1]` Semantic similarity between adversarial text and its original text.
>
> Very inspiring question! We have carried deeper investigation into this issue and please refer to `[Q1]` of the **Common Concerns** for detailed answers.
>
> > `[Q2]` The way to calculate the ASR.
>
> We totally align our evaluation protocol with prior adversarial attacks on VLP models [1,2,3], where the ASR is calculated as the proportion of successful adversarial samples within the originally correctly predicted pairs. We will supplement this introduction in the experimental part. Thanks for your kind reminder!
>
> > `[Q3]` Employment of special characters.
>
> Interesting suggestion! We employ an optimization-based strategy to obtain the text perturbation, where we iteratively update the generator to output more effective text embeddings. Actually, the vocabulary used for mapping text embeddings back to discrete words has included these special tokens such as `#` and `*`. In other words, the final adversarial word is assigned based on the optimized text embeddings and has the possibility to be a special token.
>
> To further investigate the impact of these special tokens, we conduct experiments where we directly adopt  `#` and `*` as adversarial tokens to evaluate their attack results:
>
> Table A. ASR of C-PGC and its variants using special characters as the adversarial word.
> | Source   | Target$\Rightarrow$| ALBEF |  ALBEF     | TCL   | TCL      | X-VLM  |  X-VLM     | CLIP_VIT |   CLIP_VIT    | CLIP_CNN |   CLIP_CNN    | BLIP  |   BLIP    |
> |----------|-----------|-------|-------|-------|-------|-------|-------|----------|-------|----------|-------|-------|-------|
> |    $\Downarrow$      |  Adv. word  $\Downarrow$       | TR    | IR    | TR    | IR    | TR    | IR    | TR       | IR    | TR       | IR    | TR    | IR    |
> | ALBEF    | #         | 87.81 | 85.74 | 60.84 | 62.05 | 18.28 | 35.87 | 38.67    | 61.4  | 50.91    | 68.21 | 41.71 | 54.11 |
> |   ALBEF       | *         | 87.21 | 84.87 | 60.24 | 62.19 | 18.01 | 35.4  | 38.79    | 61.91 | 51.27    | 68.1  | 41.92 | 54.69 |
> |    ALBEF      | Ours      | **90.13** | **88.82** | **62.11** | **64.48** | **20.53** | **39.38** | **43.1**     | **65.93** | **54.4**     | **72.51** | **44.79** | **56.36** |
> | CLIP_VIT | #         | 21.25 | 37.41 | 24.27 | 41.04 | 14.71 | 34.38 | 87.07    | 92.39 | 63.2     | 75.14 | 25.46 | 44.19 |
> |     CLIP_VIT     | *         | 22.07 | 37.54 | 24.58 | 41.32 | 14.81 | 34.77 | 87.57    | 92.41 | 63.78    | 74.86 | 25.76 | 44.87 |
> |    CLIP_VIT      | Ours      | **23.23** | **38.67** | **25.05** | **41.79** | **15.85** | **35.59** | **88.92**    | **93.05** | **66.06**    | **75.42** | **26.71** | **45.7**  |
>
> Table B. Comparison of C-PGC and its variants using special characters as the adversarial word regarding the BERT score between clean and adversarial texts. We adopt 5,000 texts from Flickr30 to calculate these results.
> | Source model$\Rightarrow$     | ALBEF  |   ALBEF     |   ALBEF     | TCL    |   TCL     |   TCL     | CLIP_VIT |   CLIP_VIT     |   CLIP_VIT     | CLIP_CNN |   CLIP_CNN     |   CLIP_CNN     |
> |------|--------|--------|--------|--------|--------|--------|----------|--------|--------|----------|--------|--------|
> |   Adv. word $\Downarrow$  | P$\uparrow$      | R$\uparrow$      | F1$\uparrow$     | P$\uparrow$      | R$\uparrow$      | F1$\uparrow$     | P$\uparrow$        | R$\uparrow$      | F1$\uparrow$     | P$\uparrow$        | R$\uparrow$      | F1$\uparrow$     |
> | #    | 0.8213 | 0.8419 | 0.8313 | 0.8171 | 0.8389 | 0.8277 | 0.8137   | 0.8339 | 0.8235 | 0.8156   | 0.8364 | 0.8257 |
> | *    | 0.8149 | 0.8251 | 0.8197 | 0.8098 | 0.8206 | 0.8149 | 0.8095   | 0.8206 | 0.8148 | 0.8097   | 0.8203 | 0.8147 |
> | Ours | **0.8891** | **0.8613** | **0.8748** | **0.8924** | **0.8687** | **0.8802** | **0.8746**   | **0.8684** | **0.8713** | **0.8948**   | **0.8842** | **0.8893** |
>
> Tab. A and Tab. B display that our optimization-based strategy exhibits both superior attack performance and higher semantic similarity.
> Note that the use of special tokens as adversarial words can also be more conspicuous than the natural language, which might compromise the attack stealthiness and increase the likelihood of being detected by human observers or automated filtering systems.
>
> We will supplement these discussions into the revision to promote future studies on the selection of the universal adversarial word. Thanks again for this interesting and inspiring question!

---

> ### Author Response · Authors · 2024-11-20
> **Author Response (Part II)**
>
> > `[Q4]` Details about the perturbation generator.
>
> Our generator $G_w(\cdot)$ adopts a decoder-based CNN architecture with cross-attention layers to integrate cross-modal knowledge. It upsamples a low-dimensional fixed noise vector $z_v$ into high-dimensional features. For image perturbations, $G_w(\cdot)$ directly outputs the image perturbations. For text attacks, it generates word embeddings that are subsequently flattened and mapped back to the discrete word domain.
> *i.e.*, the fundamental design principle is analogous to that of generators employed in prior studies, such as GAP [4] and Nuat-GAN [5], with differences primarily in structural composition rather than conceptual innovations.
>
> > `[Q5]` Reorganization of contributions.
>
> Thank you for your thoughtful suggestions. Our primary contributions are as follows:
>
> 1. We **design a cross-modal conditioned perturbation generator** to produce effective UAPs for both image and text modalities.
> 2. We propose **the first malicious contrastive paradigm tailored for multimodal adversarial attacks**. Firstly, we devise selection strategies (e.g., the farthest distance selection) to obtain positive and negative samples based on the attack objective. Then, we leverage these meticulously constructed samples to contrastively train the perturbation generator under both unimodal and multimodal guidance.
>
> We will more clearly reorganize and highlight the innovations and contributions in the **Introduction** sections of the paper.
>
> Thank you again for your valuable feedback! If you have any further questions or suggestions, please don’t hesitate to tell us. :)
>
> [1] Zhang J, Yi Q, Sang J. Towards adversarial attack on vision-language pre-training models. In ACM MM, 2022.
>
> [2] Lu D, Wang Z, Wang T, et al. Set-level guidance attack: Boosting adversarial transferability of vision-language pre-training models. In ICCV, 2023.
>
> [3] Wang H, Dong K, Zhu Z, et al. Transferable multimodal attack on vision-language pre-training models. In S&P, 2024.
>
> [4] Poursaeed O, Katsman I, Gao B, et al. Generative adversarial perturbations. In CVPR, 2018.
>
> [5] Gao H, Zhang H, Wang J, et al. NUAT-GAN: Generating Black-box Natural Universal Adversarial Triggers for Text Classifiers Using Generative Adversarial Networks. In IEEE TIFS, 2024.

---

### Official Review · Reviewer_jJKC · 2024-11-02

**Soundness:** 1
**Presentation:** 2
**Contribution:** 1
**Rating:** 3
**Confidence:** 5

**Summary:**

This paper proposes a method to learn universal perturbations that can transfer across different Vision-Language Pre-training (VLP) models and downstream tasks. The authors leverage contrastive loss to disrupt cross-model interactions and use a Euclidean distance-based loss to maximize the distance between adversarial data and the original data. Experimental results show that the proposed method achieves strong attack performance on various VLP models and downstream tasks.

**Strengths:**

1. The paper focuses on an important task of evaluating robustness of VLP models.
2. Both adversarial images and texts are learned.

**Weaknesses:**

Several concerns remain:
1. Motivation:
- In the abstract, the authors claim to "fully utilize the characteristics of Vision-and-Language (V+L) scenarios by incorporating both unimodal and cross-modal information." However, the authors do not seem to fully exploit the characteristics of different V+L scenarios or tasks.
- In the introduction, Figure 1 compares two methods and claims that "the generator-based approach GAP consistently achieves superior ASR compared to UAP." Since UAP uses the DeepFool method to learn perturbations, its inferior performance compared to a generator-based approach does not necessarily demonstrate the superiority of the generator-based method over other approaches, e.g., PGD. More experiments including comparisons with other strong baselines like PGD, or a more comprehensive analysis of why generator-based methods are required to support this claim.
2. Algorithm:
a: My biggest concerns are the definition of universal perturbation learning and adversarial text learning.
- The authors use generators to produce adversarial data based on cross-modal conditions. The main advantage of universal adversarial attacks is their ability to produce perturbations that are generalizable across all data without needing to generate sample-specific perturbations, thereby improving efficiency. In other words, universal perturbations should be independent of the test data and applicable to unseen data. However, relying on cross-modal conditions appears to conflict with this objective. If cross-modal conditions are required, why not generate sample-specific perturbations instead? The authors need to clarify how to maintain the universality of the perturbations without using cross-modal conditions.  Additionally, the authors should report results of universal attacks for more scenarios, e.g., using perturbations generated on the Flickr30k dataset to attack models on the MSCOCO dataset.
- Learning adversarial text perturbations requires ensuring that they do not compromise the quality of original texts. However, the authors did not address this, rendering the algorithm impractical. Further verifications are required, such as utilizing proposing metrics to evaluate the semantic consistency of perturbed texts or discussing potential methods to constrain the text perturbations to maintain readability and coherence.
b. In addition, the authors utilize contrastive learning to disrupt the cross-model relationships and use the Euclidean distance-based loss to enlarge the distance between adversarial data and their original counterpart. First, the authors utilize contrastive learning to enlarge the gap between multiple texts and minimize the distance with diverse target texts. However, I question whether setting different targets can truly maximize the distance between adversarial and original images. An ablation study is necessary to verify this, such as comparing the proposed approach with a baseline that doesn't use diverse target texts, or to analyze how different choices of target texts affect the effectiveness of the perturbations. Second, the authors use two distinct losses to maximize the distance between adversarial images and each modality in the original image-text pairs. It is unclear why two losses are needed, rather than using a unified loss for both modalities. Additional experiments comparing their two-loss approach with a unified loss approach, and analyze the impact on both cross-modal and intra-modal relationships should be conducted.

c. Perturbation learning methods, including set-level augmentation, maximizing both intra- and inter-model differences, and leveraging contrastive learning, have already been explored by current approaches [1,2]. The specific contribution of this method remains unclear, aside from generating universal perturbations instead of sample-specific ones. Furthermore, the distinction between the generation of universal and sample-specific perturbations remains unclear.

5. Experiments:
-  Previous works on universal adversarial attacks for VLP models should be discussed and compared, such as [3]. Additional comparison with relevant methods should be provided.
- The authors apply data augmentation to improve the method's effectiveness, but additional comparisons with other augmentation techniques should be conducted to better demonstrate the proposed method's superiority. Examples include ScMix [3] and Admix [4].
 [1] Dong Lu, Zhiqiang Wang, Teng Wang, Weili Guan, Hongchang Gao, and Feng Zheng. 2023. Set-level guidance attack: boosting adversarial transferability of vision-language pre-training models. In Proceedings of the IEEE International Conference on Computer Vision. 102–111.
[2] Kim M, Tack J, Hwang S J. Adversarial self-supervised contrastive learning[J]. Advances in neural information processing systems, 2020, 33: 2983-2994.
[3] Xiaosen Wang, Xuanran He, Jingdong Wang, and Kun He. 2021. Admix: enhancing the transferability of adversarial attacks. In Proceedings of the IEEE International Conference on Computer Vision. 16158–16167.
[4] Zhang, Peng-Fei, Zi Huang, and Guangdong Bai. "Universal Adversarial Perturbations for Vision-Language Pre-trained Models." Proceedings of the 47th International ACM SIGIR Conference on Research and Development in Information Retrieval. 2024.

**Questions:**

Please refer to Weaknesses.

---

> ### Author Response · Authors · 2024-11-20
> **Author Response (Part I)**
>
> We sincerely thank you for your precious time and effort in providing a wealth of suggestions to enhance the quality of our paper. We have carefully read all the comments and provide detailed point-by-point responses as follows. Hopefully, we can adequately address your concerns.
>
> > `[Q1.1]` Fully utilize the characteristics of V+L scenario.
>
> Thanks for your constructive advice. This claim aims to highlight that in contrast to the **unimodal scenarios**, we fully leverage the unique characteristics of **multimodal scenarios** to enhance the modeling of a universal perturbation that can generalize to various samples and V+L tasks.
> Specifically, we elaborate on this claim from the following main aspects.
>
> 1. **Cross-Modal Conditions in Perturbation Generator**:
>    We design a perturbation generator with cross-modal conditions to benefit from cross-modal knowledge. In contrast, generators in unimodal generative attacks rely solely on information from a single modality.
> 2. **Malicious Multimodal Contrastive Learning Paradigm**:
>    We formulate a multimodal contrastive learning paradigm for multimodal adversarial attacks. In comparison, unimodal attacks typically focus on interactions within a single modality to generate adversarial samples.
> 3. **Joint Perturbations Across Modalities**:
>    Similar to prior sample-specific adversarial attacks on VLP models [1, 2], our approach applies perturbations to both image and text modals. Conversely, unimodal attacks only allow the perturbation of a single modality.
>
>  Extensive experiments demonstrate the effectiveness of these techniques, revealing that our framework utilizes the characteristics of V+L scenarios to achieve superior performance.
>
> Besides, while investigating task-level V+L characteristics to enhance attacks for specific tasks is a promising direction, we highlight that the primary focus of our paper lies in **leveraging the shared and joint characteristics of Vision-Language scenarios** to present a universal and versatile UAP that can effectively generalize to **diverse downstream V+L tasks**.
> Thanks again for your thoughtful comment. We will supplement these analyses in **Appendix I** of our revision.
> > `[Q1.2]` Comparison with PGD-based UAP algorithm.
>
>  Thanks for this insightful suggestion!
> Following your advice, we supplement the comparison between PGD-based UAP and GAP as follows：
>
> Table A. ASR results of GAP and UAP learned by DeepFool and PGD respectively.
> | Source$\Rightarrow$ | ALBEF | ALBEF      | TCL   |   TCL    | X-VLM |  X-VLM     | CLIP_VIT | CLIP_VIT      | CLIP_CNN |  CLIP_CNN     | BLIP  |    BLIP   |
> |---------|-------|-------|-------|-------|------|-------|----------|-------|----------|-------|-------|-------|
> |   Method$\Downarrow$    | TR    | IR    | TR    | IR    | TR   | IR    | TR       | IR    | TR       | IR    | TR    | IR    |
> | UAP_DeepFool     | 13.98 | 14.95 | 12.28 | 16.59 | 3.66 | 13.78 | 20.94    | 36.68 | 26.42    | 41.88 | 11.85 | 18.76 |
> | UAP_PGD | 19.83 | 16.88 | 15.8  | 18.54 | 4.25 | 14.08 | 20.41    | 35.58 | 27.52    | 42.7  | 12.73 | 18.98 |
> | GAP     | **69.78** | **81.59** | **22.15** | **29.97** | **6.61** | **18.37** | **23.4**     | **37.54** | **29.92**    | **44.29** | **16.09** | **28.12** |
>
> The results indicate that the UAP learned through PGD indeed obtains better results compared to the DeepFool-based one. Nevertheless, GAP still achieves significantly higher fooling rates, validating the necessity of employing a generator.
> Furthermore, numerous studies on generative adversarial attacks [3,4,5] have also similarly highlighted the superior performance of generative methods over non-generative ones, thereby supplementing strong empirical support for our claim.
>
> We will include these experimental results and cite the related literature in the **Introduction** section to more confidently affirm our conclusion: *"The generator-based approach GAP consistently achieves superior ASR compared to UAP."* Thanks again for your valuable suggestion that helps us improve our paper.

---

> > ### Comment · Reviewer_jJKC · 2024-11-26
> > **Response to Author Response (Part I)**
> >
> > Thanks for authors' efforts for address the reviewer's concerns. I have several questions need to be further addressed.
> >
> > 1. In **[Q1.1]**, leveraging cross-modal knowledge, such as the paired relationships between images and texts, is not a novel concept in cross-modal attacks. Are there any other new insights derived from this task? Furthermore, the authors claim to fully utilize the characteristics of vision-and-language (V+L) scenarios. However, different cross-modal tasks—such as cross-modal retrieval and image captioning—focus on distinct contents and relationships between images and texts, each with unique characteristics. It appears that the authors have not sufficiently explored these task-specific characteristics, raising concerns that the approach may not fully leverage the diverse aspects of V+L scenarios.
> >
> > 2. In **[Q1.2]**, why do generative methods achieve significantly better performance compared to PGD methods? Is there any underlying rationale behind this difference?

---

> ### Author Response · Authors · 2024-11-20
> **Author Response (Part II)**
>
> > `[Q2.1]` Definition of universal perturbation learning
>
>   We are deeply sorry for any misunderstanding our paper may cause you. Please kindly allow us to restate the training and inference paradigm of C-PGC to clarify the use of cross-modal conditions.
>
> Taking image perturbation as an example, the attacker has access to a training dataset of image-text pairs to learn a perturbation generator that transforms **a fixed noise** $z_v$ into the universal perturbation $\delta_{v}$.
> Specifically, for each training image-text pair $(v, \mathbf{t})$, the texts $\\mathbf{t}$ are fed into the generator through cross-attention layers as cross-modal conditions, to provide additional knowledge for learning an effective UAP.
>
> However, once we finish the contrastive training of the generator, the final output $\delta_{v}$ becomes the image UAP that can then **be applied to any test data without further requiring any cross-modal conditions**.
> This is consistent with established practices of universal perturbation [6,7,8], and ensures the universality and applicability of the universal perturbation, which is definitely independent of the test data and unseen data.
>
> Also, our experimental setup fully aligns with previous UAP studies [6, 8, 9] and uses the training set of datasets (e.g., MSCOCO) to learn the UAP while evaluating the validation set. Moreover, experiments involving *“using perturbations generated on the Flickr30k dataset to attack models on the MSCOCO dataset”* actually **have been presented in Table 7 of our Appendix** (Table 12 in the revised version). We apologize for the lack of emphasis on this matter and will highlight this experiment in the main text.
>
> > `[Q2.2.1]` Semantic similarity between adversarial text and its original text
>
> Thank you for this meaningful question. Please kindly refer to `[Q1]` in the **Common Concerns** for a detailed response.
>
> > `[Q2.2.2]` Ablation study of diverse target texts.
>
> Very insightful suggestion! Following your advice, we implement a variant C-PGC${_{Sin}}$, which uses only a single target text with the farthest distance as the positive sample:
>
> Table B. Comparison of C-PGC and its variant with single text as the target.
> | Source   |Target$\Rightarrow$ | ALBEF | ALBEF      | TCL   |  TCL     | X-VLM  |  X-VLM     | CLIP_VIT |  CLIP_VIT     | CLIP_CNN |   CLIP_CNN    | BLIP  |    BLIP   |
> |----------|----------|-------|-------|-------|-------|-------|-------|----------|-------|----------|-------|-------|-------|
> |     $\Downarrow$     |   Method$\Downarrow$       | TR    | IR    | TR    | IR    | TR    | IR    | TR       | IR    | TR       | IR    | TR    | IR    |
> | ALBEF    | C-PGC${_{Sin}}$ | 82.99 | 86.14 | 49    | 56.98 | 18.19 | 35.79 | 40.52    | 65.9  | 51.09    | 69.68 | 38.54 | 52.86 |
> |    ALBEF      | Ours     | **90.13** | **88.82** | **62.11** | **64.48** | **20.53** | **39.38** | **43.1**     | **65.93** | **54.4**     | **72.51** | **44.79** | **56.36** |
> | CLIP_VIT | C-PGC${_{Sin}}$ | 20.55 | 37.46 | 24.43 | 41.39 | 13.52 | 32.6  | 79.93    | 88.64 | 55.44    | 69.43 | 24.4  | 43.06 |
> |    CLIP_VIT      | Ours     | **23.23** | **38.67** | **25.05** | **41.79** | **15.85** | **35.59** | **88.92**    | **93.05** | **66.06**    | **75.42** | **26.71** | **45.7**  |
>
> The results illustrate that the use of multiple target texts can enhance attack effectiveness, validating the efficacy of set-level diverse guidance.
> As for the choices of target texts, we have provided **an alternative strategy C-PGC$_{Rand}$ in Section 4.4 of the main text**, where target texts are randomly selected instead of choosing the farthest ones. This finding verifies the effectiveness of our farthest selection strategy.
>
> We will supplement the experiment of C-PGC$_{Sin}$ into the **Section E** of the Appendix. We appreciate your valuable suggestion for improving the comprehensiveness of our analysis.

---

> > ### Comment · Reviewer_jJKC · 2024-11-26
> > **Response to Author Response (Part II)**
> >
> > 3. **[Q2.2.1]** In the second question from the original reviewer, my concern is how to make sure that adversarial perturbations on texts would not influence the quality of original texts.  Calculating the semantic similarity between clean and adversarial texts cannot solve this problem. Instead, the higher similarity between them show that the proposed method does not significantly alter the original semantics in the feature spaces. Thus, it cannot achieve effective attacks. The authors should prove that adversarial texts are visually plausible.

---

> ### Author Response · Authors · 2024-11-20
> **Author Response (Part III)**
>
> > `[Q2.2.3]` Investigation of the loss function. (Updated)
>
> Inspiring question! The two designed loss terms $\\mathcal{L}\_{CL}$ and $ \\mathcal{L}\_{Dis}$ serve distinct roles in reaching the attack goal from multimodal and unimodal perspectives respectively. We have conducted **ablation studies in Section 4.4 of the main body** and corroborated the effectiveness of both two terms, hence confirming the necessity of jointly employing both $\\mathcal{L}\_{CL}$ and the $ \\mathcal{L}\_{Dis}$.
>
> To further address your concerns, we conduct experiments with different unified loss alternatives C-PGC$\_{Cos}$ and C-PGC$\_{MSE}$, which maximizes the negative cosine similarity and MSE between features of matched image-text pairs.
> Table C. ASR of C-PGC and its three variants on Flickr30. The surrogate model is ALBEF.
> | Target$\Rightarrow$| ALBEF | ALBEF      | TCL   |   TCL    | X-VLM  | X-VLM      | CLIP_VIT |  CLIP_VIT     | CLIP_CNN |   CLIP_CNN    | BLIP  |    BLIP   |
> |----------|-------|-------|-------|-------|-------|-------|----------|-------|----------|-------|-------|-------|
> |   Method$\Downarrow$       | TR    | IR    | TR    | IR    | TR    | IR    | TR       | IR    | TR       | IR    | TR    | IR    |
> | C-PGC$\_{MSE}$      | 12.02 | 30.75 | 14.39 | 35.08 | 11.41 | 30.79 | 37.32    | 56.05 | 40.17    | 56.39 | 19.66 | 37.33 |
> | C-PGC$\_{Cos}$      | 57.55 | 67.4  | 37.06 | 49.45 | 10.7  | 28.48 | 37.49    | 58.3  | 40.87    | 58.39 | 23.33 | 39.44 |
> | Ours     | **90.13** | **88.82** | **62.11** | **64.48** | **20.53** | **39.38** | **43.1**     | **65.93** | **54.4**     | **72.51** | **44.79** | **56.36** |
>
> These results again reveal the superiority of the proposed loss function over possible unified alternatives, which will also be added in **Section E** of the Appendix.
>
> > `[Q2.3]` Contribution of our work.
>
> (1) As stated in the main text, the set-level augmentation was proposed by SGA [2] and we incorporate it into our framework for further enhancement, without claiming it as the major contribution.
> (2) Meanwhile, it is reasonable and common practice to pursue a strategy that jointly maximizes both types of diversity. However, the pivotal distinction lies in how it is effectively implemented and the degree to which it enhances attack efficacy.
> (3) We also highlight that contrastive learning is a general concept that has indeed been utilized by previous approaches for general beneficial purposes, such as modality alignment and enhancing adversarial robustness [10].
>
> However, our contribution lies in making **the first attempt to propose a novel contrastive paradigm tailored to the malicious scenario of vision-language UAP generation**, which effectively fools different VLP models across diverse V+L scenarios. Based on the attack objective, we devise **tailored positive and negative sample selection strategies** (*e.g.*, the farthest distance selection) to facilitate the contrastive learning of UAP. Moreover, we propose **a novel architecture of perturbation generator with cross-modal conditions** as auxiliary information to promote the modeling of multimodal UAP. By training the generator with both the introduced **set-level cross-modal ($\mathcal{L}\_{CL}$) and unimodal ($\mathcal{L}\_{Dis}$) guidance**, C-PGC generates powerful UAP against VLP models with excellent generalization and transferability.
>
> As for the distinction between universal and sample-specific attacks, we hope that our response in `[Q2.1]` has adequately solved your concerns.
>
> > `[Q3.1]` Comparison with a recent baseline.
>
> Thank you for your constructive suggestion! I guess you are referring to [4] in your cited references, is that correct? We have supplemented our comparison with this excellent concurrent work and please refer to `[Q2]` in the **Common Concerns** for details.

---

> ### Author Response · Authors · 2024-11-20
> **Author Response (Part IV)**
>
> > `[Q3.2]` Comparison of different data augmentations.
>
> Thank you for this valuable suggestion. Initially, we were motivated by the impressive performance of SGA [2] and hence integrated their image augmentation into our framework to enhance the attack.
> Following your advice, we reproduce your suggested augmentation techniques while keeping other settings unchanged，*i.e.*, C-PGC$\_{ScMix}$ and C-PGC$\_{Admix}$. Experimental results are presented as follows:
>
> Table D. Attack performance under different data augmentation strategies.
> | Source   |Target$\Rightarrow$ | ALBEF |  ALBEF     | TCL   |  TCL     | X-VLM  |    X-VLM   | CLIP_VIT |   CLIP_VIT    | CLIP_CNN |  CLIP_CNN     | BLIP  |   BLIP    |
> |----------|----------|-------|-------|-------|-------|-------|-------|----------|-------|----------|-------|-------|-------|
> |   $\Downarrow$       | Strategy$\Downarrow$      | TR    | IR    | TR    | IR    | TR    | IR    | TR       | IR    | TR       | IR    | TR    | IR    |
> | ALBEF    | C-PGC$\_{ScMix}$    | 66.08 | 76.26 | 39.03 | 51.24 | 20.73 | 37.47 | 40.02    | 65.58 | 50.13    | 71.85 | 34.6  | 51.9  |
> |    ALBEF      | C-PGC$\_{Admix}$   | 62.8  | 72.23 | 34.47 | 47.78 | 19    | 36.67 | 42       | 64.88 | 48.19    | 69.68 | 32.28 | 50.05 |
> |     ALBEF     | Ours     | **90.13** | **88.82** | **62.11** | **64.48** | **20.53** | **39.38** | **43.1**     | **65.93** | **54.4**     | **72.51** | **44.79** | **56.36** |
> | CLIP_VIT | C-PGC$\_{ScMix}$    | 20.55 | 37.46 | 24.43 | 41.39 | 13.52 | 32.6  | 79.93    | 88.64 | 55.44    | 69.43 | 24.4  | 43.06 |
> |    CLIP_VIT      | C-PGC$\_{Admix}$    | 19.53 | 37.04 | 24.02 | 41.5  | 14.74 | 34.26 | 85.34    | 91.8  | 59.07    | 71.78 | 23.66 | 43.22 |
> |    CLIP_VIT      | Ours     | **23.23** | **38.67** | **25.05** | **41.79** | **15.85** | **35.59** | **88.92**    | **93.05** | **66.06**    | **75.42** | **26.71** | **45.7**  |
>
> It can be observed that C-PGC with the current augmentation strategy outperforms the ScMix and Admix, hence validating the set-level guidance is more suitable for our contrastive training. This is achieved by SGA's alignment-preserving augmentation, which enriches image-text pairs while maintaining their inherent alignments intact [2], hence better maintaining the effectiveness of our malicious contrastive learning. We will include these results in **Section E** of the Appendix to provide insights for future research.
>
> Finally, we would like to express our gratitude once again for your perceptive and valuable feedback. We hope that the comprehensive response above will effectively address your concerns. It would be our great pleasure to engage in further discussion with you.
>
> [1] Zhang J, Yi Q, Sang J. Towards adversarial attack on vision-language pre-training models. In ACM MM, 2022.
>
> [2] Lu D, Wang Z, Wang T, et al. Set-level guidance attack: Boosting adversarial transferability of vision-language pre-training models. In ICCV, 2023.
>
> [3] Hayes J, Danezis G. Learning universal adversarial perturbations with generative models. In IEEE Security and Privacy Workshops (SPW), 2018.
>
> [4] Feng W, Xu N, Zhang T, et al. Dynamic generative targeted attacks with pattern injection. In CVPR, 2023.
>
> [5] Gao H, Zhang H, Wang J, et al. NUAT-GAN: Generating Black-box Natural Universal Adversarial Triggers for Text Classifiers Using Generative Adversarial Networks. In IEEE TIFS, 2024.
>
> [6] Moosavi-Dezfooli S M, Fawzi A, Fawzi O, et al. Universal adversarial perturbations. In CVPR, 2017.
>
> [7] Chaubey A, Agrawal N, Barnwal K, et al. Universal adversarial perturbations: A survey. In arXiv, 2020.
>
> [8] Zhang, Peng-Fei, Zi Huang, and Guangdong Bai. Universal Adversarial Perturbations for Vision-Language Pre-trained Models. In ACM SIGIR, 2024.
>
> [9] Poursaeed O, Katsman I, Gao B, et al. Generative adversarial perturbations. In CVPR, 2018.
>
> [10] Kim M, Tack J, Hwang S J. Adversarial self-supervised contrastive learning. In NIPS, 2020.

---

> ### Author Response · Authors · 2024-11-23
> **A Kind Reminder of the Author Response**
>
> Dear Reviewer jJKC,
>
> Thank you again for your precious efforts in reviewing our paper and the constructive comments!
>
> As the end of the discussion period approaches, we would like to know whether our responses have properly addressed your concerns. Your feedback will be highly appreciated and we are glad to engage in further discussions with you.
>
> Sincerely,
>
> The Authors

---

> ### Author Response · Authors · 2024-11-26
> **Looking Forward to More Discussions**
>
> Dear Reviewer jJKC,
>
> We have carefully responded to each of your questions and revised our paper accordingly. The sufficient experiments and analyses have been presented above and the revision details are listed at the top of this page.
>
> We look forward to your reply and welcome discussion on any issues regarding our paper and responses.
>
> Best regards,
>
> The Authors

---

> ### Author Response · Authors · 2024-11-26
> **Author Response (Part V)**
>
> Thanks for your detailed feedback! Your acknowledgment of our above responses encourages us a lot. Next, we provide point-to-point responses below to further resolve your concerns.
>
> > `[Q1.1]` Clarification about Leveraging Cross-modal Knowledge.
>
> A very insightful question! We highlight that the use of cross-modal knowledge in previous cross-modal attacks is **limited to simply maximizing the feature distance between samples of different modals for adversarial sample optimization**, without any deeper utilization to further enhance attacks [1,2,8].
>
> On the contrary, we are **the first to incorporate cross-modal information into the perturbation generator as auxiliary knowledge to facilitate the modeling of multimodal adversarial perturbation**.
> Moreover, we propose **the first attack paradigm that utilizes cross-modal knowledge from a novel perspective of malicious contrastive learning**. Extensive experimental results have validated the effectiveness of our proposed novel techniques to utilize the cross-modal knowledge, presenting a powerful universal attack framework.
>
> > `[Q1.2]` Task-specific characteristics
>
> We are deeply sorry for any misunderstanding our response may cause you.
>
> We clarify that the essential attack objective of this work aligns with existing well-acknowledged adversarial attacks on VLP models [1,2,8], which **aim to fool VLP models themselves, rather than specific downstream tasks**. *i.e.*, the generated perturbation is supposed to **yield excellent attack performance tailored to the target VLP model regardless of the downstream tasks**. Hence, our primary focus is to leverage **the shared and joint characteristics of different Vision-Language scenarios** to present a universal and versatile UAP that can effectively generalize to diverse downstream V+L tasks.
>
> Also, it is a promising direction for future studies to investigate task-level V+L characteristics to further enhance attacks for specific downstream tasks. We have added them to encourage future studies in **Appendix I** of the revision.
>
> > `[Q2]` The Superiority of Generative Attacks
>
> We're sorry for lacking a sufficient explanation regarding the underlying mechanism behind the generative attack.
>
> The superiority is achieved by the **powerful distributional modeling capability of generative model**. Since a universal perturbation is learned on the data distribution independent of specific instances, the generator facilitates the learning of universal perturbation by better perceiving and capturing distribution-level features of diverse image-text samples  [4,9,11,12], hence significantly enhancing the attack compared with non-generative methods.

---

> ### Author Response · Authors · 2024-11-26
> **Author Response (Part VI)**
>
> > `[Q3]` Influence the quality of original texts.
>
> We sincerely apologize for the misunderstanding our response may cause you. Please kindly allow us to provide a more detailed explanation of this issue as follows.
>
> **1. Clarifications of the attack goal.**
>
> (1) We clarify that ensuring the quality of the initial text is not compromised is indeed valuable, but the basic objective of a malicious untargeted adversarial attacker is to **fool the victim model to output incorrect predictions while ensuring attack imperceptibility [1, 2, 13, 14], rather than not influence the quality of original data**.
>
> (2) Correspondingly, our **attack performance** is verified by the excellent ASR of extensive experiments **in the main text** and the comparison with your kindly suggested baseline **in Table C of the Common Concerns**, and the **attack stealthiness** is guaranteed by the results measured by sufficient distance metrics **in Table A and B of the Common Concerns**, presenting a qualified and successful untargeted adversarial attack method.
>
> **2. Evaluation of the influence on the quality of original texts.**
>
> (3) We emphasize that how to evaluate and reduce the extent of influencing the quality of original texts is still **an open issue that has not been explored by existing cross-modal adversarial attacks [1,2,8,15]**.
>
> Since the original text is the reference representing the original sentence semantics, the semantic similarity between the adversarial and original texts measured by sufficient text distance metrics **can be regarded as a reasonable metric to evaluate the extent of influence on the text quality**. Therefore, the results in **Tables A and B of the Common Concerns** verify C-PGC's better performance in maintaining text quality compared with existing well-acknowledged methods [1,2].
>
> (4) To further address your concerns, we provide several visual demonstrations between our method and SGA [4]. We observe that, in most cases, our method indeed exerts less impact on the quality of original texts.
>
>
>     (Clean) Man taking a photograph of a well-dressed group of teens.
>     (SGA) Man [rights] a photograph of a well-dressed group of teens.
>     (Ours) Man [getting] a photograph of a well-dressed group of teens.
>
>     (Clean) A young girl wearing a bulky red life jacket floating in a lake.
>     (SGA) A young girl wearing a bulky red life [school] floating in a lake.
>     (Ours) A young girl [getting] a bulky red life jacket floating in a lake.
>
>     (Clean) A brown dog walks in the grass with its tongue hanging out.
>     (SGA) A brown dog walks in the [new] with its tongue hanging out.
>     (Ours) A brown dog [getting] in the grass with its tongue hanging out.
>
>     (Clean) Two young men are loading fruit onto a bicycle.
>     (SGA) [Teens] young men are loading fruit onto a bicycle.
>     (Ours) Two young men are [getting] fruit unto a bicycle.
>
>     (Clean) A dog is walking through some gravel beside a river.
>     (SGA) A dog is walking through some [like] beside a river.
>     (Ours) A dog is [getting] through some gravel beside a river.
> These visualization results correspond with our numeric results in **Table A and B of the Common Concerns**. More visualization results will be added to the Appendix. Thanks again for your valuable question!
>
> We're more than glad to have more discussions with you if you have any further questions. :)
>
>
> [11] Naseer M M, Khan S H, Khan M H, et al. Cross-domain transferability of adversarial perturbations. In NIPS, 2019.
>
> [12] Yang X, Dong Y, Pang T, et al. Boosting transferability of targeted adversarial examples via hierarchical generative networks. In ECCV 2022.
>
> [13] Dong Y, Liao F, Pang T, et al. Boosting adversarial attacks with momentum. In CVPR, 2018.
>
> [14] Chakraborty A., Alam M., Dey V., et al. A survey on adversarial attacks and defences. In CAAI Transactions on Intelligence Technology, 2021.
>
> [15] Wang H, Dong K, Zhu Z, et al. Transferable multimodal attack on vision-language pre-training models. In S&P, 2024.

---

> ### Author Response · Authors · 2024-11-28
> **A Gentle Reminder of the Latest Author Response**
>
> Dear Reviewer jJKC,
>
> We really appreciate your precious time and efforts in providing insightful feedback on our responses.
> To further address your concerns, we have presented detailed clarifications and visual demonstrations in the latest author's reply.
> We kindly request that you review them to see whether your remaining queries have been sufficiently addressed. If you are satisfied with our responses, we would greatly appreciate it if you could kindly consider raising your score accordingly. :)
>
> We're looking forward to your reply.
>
> Best regards,
>
> Authors

---

> ### Author Response · Authors · 2024-12-01
> **Author Response (Part VII)**
>
> > Supplementary experiment to '`[Q3]` Influence the quality of original texts'
>
> To better solve your question, we devise a novel experiment **leveraging the advanced GPT-4o** to quantitatively evaluate the influence of text perturbation on the quality of the original sentence. Specifically, we query GPT-4o with the following prompt inspired by [16], using 5000 pairs of original and perturbed texts.
>
>     As an experienced evaluator, your task is to evaluate the extent of the given word replacement operation to the original text quality. Give a specific integer score based on the following statements, ranging from 0 to 4: Very Poor (0): The word replacement greatly impacts the quality of the initial sentence, e.g., largely destroying the basic semantic or basic grammar.
>     Poor (1): The word replacement causes significant semantic deviation and ambiguity, e.g., the perturbed sentence looks strange and illogical.
>     Fair (2): The perturbed sentence is readable and logical, showing likeness to the initial sentence with notable semantic variances.
>     Good (3): The perturbed sentence is reasonable and semantic-preserving.
>     Excellent (4): The word replacement nearly brings no significant influence on the sentence quality.
>
>     Every time you receive two sentences, the first one is an original sentence, and the second is a sentence perturbed by the word replacement operation.
>
> The obtained average scores are as follows:
>
> Table E. GPT-4o score of different methods in terms of the influence on the text quality. The high values indicate better performance.
>
> | Eval Model | Co-Attack | SGA   | C-PGC  |
> |------------|-----------|-------|--------|
> | GPT-4o     | 2.6846    | 2.802 | **2.9164** |
>
> These results again prove that C-PGC achieves better performance in terms of the influence on the original text quality. Among existing cross-modal attacks [1,2,8,15], we make the first attempt to quantitatively evaluate the direct influence of text perturbation on the original sentence quality based on this experiment, which serves as an important addition to better solve your question.
>
> [16] Peng Y, Cui Y, Tang H, et al. Dreambench++: A human-aligned benchmark for personalized image generation. In arXiv, 2024.

---

> ### Author Response · Authors · 2024-12-02
> **A Gentle Reminder of the End of Rebuttal**
>
> Dear Reviewer jJKC,
>
> This is a kind reminder that the end of the rebuttal is really close. We have addressed the majority of your concerns in our first-stage responses and supplemented further experiments with detailed analyses of your remaining issues. If you're satisfied with our rebuttal, we would appreciate it if you could consider raising your score accordingly. :)
>
> Best regards,
>
> Authors

---

> > ### Comment · Reviewer_jJKC · 2024-12-02
> > **Response to the Authors**
> >
> > Thank you for your response. My feedback is as follows:
> >
> > Re: [Q1.1] Clarification about Leveraging Cross-modal Knowledge
> >
> > I suggest that the authors reflect this point in the revised version of the paper, as it is crucial to highlight the differences between attacks targeting Vision-Language Pre-training Models and those designed for other types of models.
> >
> > Re: [Q3] Influence on the Quality of Original Texts
> >
> > I still find the similarity measurement problematic. The goal of an attack is to increase the gap between adversarial data and the original data. If the semantic similarity between them is low, the attack becomes less effective. A few examples provided are insufficient to confirm that the adversarial texts are visually plausible.
> >
> > Considering the concern regarding the adversarial text, I will maintain my score.

---

> ### Author Response · Authors · 2024-12-02
> **Author Response (Part VIII)**
>
> > Re to "Re: [Q1.1] Clarification about Leveraging Cross-modal Knowledge"
>
> Sure, we will incorporate this into the **introduction and method** part of this paper.
>
> >  Re to "Re: [Q3] Influence on the Quality of Original Texts"
>
> We're quite confused about your subjective opinion regarding the similarity measurement after our detailed clarification and experimental results.
>
> 1. The attack performance of our method has been directly measured by the ASR of extensive experimental results. We have highlighted them again and provided a comparison with your suggested SOTA baseline UAP method, **which strongly confirms the superiority of our method**. It's unclear why you insist on the similarity measurement to doubt the attack performance.
> 2. We have emphasized that **ensuring the quality of the initial text is not the basic objective of adversarial attacks**, but attack stealthiness. You may mistakenly mix them up and request harsh constraints on the text perturbation that **almost all the existing cross-modal attacks fail to satisfy**.
> 3. We have highlighted the evaluation and reduction of the extent of influencing the quality of original texts is **still an open issue that has not been explored** by existing cross-modal adversarial attacks [1,2,8,15].
> 4. We also aim to convey that existing instance-specific attacks exhibit great attack performance due to their instance-wise adversarial perturbation, while they **sacrifice the semantic similarity and lead to less stealthy attacks (also compromise the text quality that you're most concerned about)**. Since our method achieves both better semantic similarity and scores of influence on the text quality than existing instance-specific attacks, C-PGC is a highly qualified adversarial attack method and is definitely acceptable in terms of attack stealthiness and influence on the text quality.
>
> Despite the above facts, we still design and provide sufficient new experiments to solve your concerns (**including the direct analysis with the advanced GPT-4o**), which repeatively validate the better quality-preserving of our method. However, you seem to neglect the supplemented experiments and keep holding subjective opinions about the measurement, **without specifying any possible useful metrics or specific problems within the evaluation protocol**.
>
> We kindly request you review our provided experiments and re-assess the quality of this paper. This is the true meaning of the rebuttal phrase of ICLR. And your precious assessment also means a lot to us.

---

> ### Author Response · Authors · 2024-12-02
> **Appeal to Responses to the Latest Author Response**
>
> Dear Reviewer jJKC,
>
> Sorry for reminding you again since the end of the rebuttal is really close. You seem to neglect our latest quantitative experiments based on GPT-4o in **Author Response (Part VII)**. We also present detailed clarification regarding your confusing reasons for not increasing scores in **Author Response (Part VIII)**.
>
> We have made great efforts to solve your issues and would appreciate a more fair and objective evaluation.
>
> Best regards,
>
> Authors

---

### Author Response · Authors · 2024-11-20
**Common Concerns (Part I)**

We sincerely appreciate all the reviewers for dedicating their valuable time and effort to review our paper and provide insightful comments and suggestions.
Encouragingly, reviewers praise C-PGC’s design (`R#U4zG`, `R#AoYQ`), effectiveness (`R#AoYQ`, `R#U4zG`), scalability and versatility (`R#AoYQ`,`R#U4zG`) and appreciate the writing (`R#U4zG`, `R#4NRc`) and experiments (`R#AoYQ`, `R#U4zG`, `R#4NRc`) of our work. Below, we uniformly address several concerns shared by different reviewers.

> `[Q1]` **R#jJKC**, **R#AoYQ**: Semantic Similarity between Clean and Adversarial Texts.

Thanks for the insightful question. Yes! The basic objective of untargeted adversarial attacks is to fool the victim model to output incorrect predictions [1, 2], while the attacker is supposed to preserve semantic similarity between the original and the adversarial sample to **ensure attack imperceptibility**.
In our implementation, we follow the rigorous setup in prior works [3,4,5] that modify only one single word to preserve semantic similarity and attack stealthiness.
To further quantitatively evaluate C-PGC, we follow the suggestion of **R#AoYQ** and **calculate BERT scores [6], including P (precision), R (recall), and F1 (F1 score) as metrics** for the semantic distance between 5,000 clean and adversarial sentences in Table A. To more comprehensively evaluate, we also compute BLEU metrics when surrogated on ALBEF in Table B.

Table A. Comparison of our C-PGC with the two widely acknowledged Co-Attack [3] and SGA [4] regarding the semantic similarity between clean and adversarial texts.
| Source model$\Rightarrow$   | ALBEF  |   ALBEF     |   ALBEF     | TCL    |  TCL      |   TCL     | CLIP_VIT |   CLIP_VIT     |  CLIP_VIT      | CLIP_CNN |   CLIP_CNN     |  CLIP_CNN      |
|-----------|--------|--------|--------|--------|--------|--------|----------|--------|--------|----------|--------|--------|
|     Method$\Downarrow$      | P$\uparrow$    | R$\uparrow$        | F1$\uparrow$       | P  $\uparrow$        | R $\uparrow$         | F1$\uparrow$         | P$\uparrow$            | R$\uparrow$          | F1$\uparrow$         | P$\uparrow$            | R$\uparrow$          | F1$\uparrow$         |
| Co-Attack [3] | 0.8328 | 0.8589 | 0.8455 | 0.8325 | 0.8588 | 0.8453 | 0.8269   | 0.8526 | 0.8394 | 0.8271   | 0.853  | 0.8397 |
| SGA [4]      | 0.8389 | **0.8654** | 0.8518 | 0.8376 | 0.8646 | 0.8509 | 0.8416   | **0.8697** | 0.8553 | 0.8378   | 0.865  | 0.8511 |
| Ours      | **0.8891** | 0.8613 | **0.8748** | **0.8924** | **0.8687** | **0.8802** | **0.8746**   | 0.8684 | **0.8713** | **0.8948**  | **0.8842** | **0.8893** |

Table B. Comparison of BLEU metrics regarding our C-PGC with Co-Attack and SGA.
| Method $\Downarrow$    | B@4   | METEOR | ROUBE_L | CIDEr | SPICE |
|-----------|-------|--------|---------|-------|-------|
| Co-Attack [1] | 0.79  | 0.52   | 0.895   | 7.03  | 0.661 |
| SGA [2]      | 0.798 | 0.527  | 0.898   | 7.159 | 0.668 |
| Ours      | **0.889** | **0.552**  | **0.905**   | **8.036** | **0.671** |

Note that we provide previous sample-specific algorithms `Co-Attack`[3] and `SGA`[4] as references.
Notably, our method  **achieves better similarity scores to these wide-acknowledged sample-specific methods** across different surrogate VLP models, demonstrating the outstanding attack stealthiness of our C-PGC.

To better understand its influence on attack performance under more practical scenarios, we evaluate C-PGC against a widely used language correction tool, *i.e*., the **LangugeTool** (LT) which has been applied to adversarial text correction [5]. As shown in **Table 2 of the main body**, the minor ASR drop from NRP to NRP+LT indicates that C-PGC exhibits excellent attack imperceptibility and can effectively bypass the automated evaluation tool, which again underscores the generalizability of the generated UAP, presenting a practical and effective textual perturbation strategy.

We will supplement these analyses in the Appendix and encourage more related future works. Thanks again for your insightful advice!

---

> ### Author Response · Authors · 2024-11-20
> **Common Concerns (Part II)**
>
> > `[Q2]` **R#jJKC**, **R#U4zG**: Comparison with the recent SOTA.
>
> As noted by **R#jJKC**, we notice a concurrent study on UAP attacks for VLP models [7], which shows promising attack performance.
> To make a fair comparison, we faithfully reproduce this method using their publicly released code under the same experimental settings as ours. Note that [7] implements several versions of their method and we report their best results as follows:
>
> Table C. Comparison of our C-PGC with a recent SOTA attack [7] on Flicke30K.
> | Source   |Target$\Rightarrow$     | ALBEF |  ALBEF     | TCL   |   TCL    | X-VLM  |  X-VLM     | CLIP_VIT |    CLIP_VIT   | CLIP_CNN |  CLIP_CNN     | BLIP  |  BLIP     |
> |----------|------------|-------|-------|-------|-------|-------|-------|----------|-------|----------|-------|-------|-------|
> |    $\Downarrow$        |    Method$\Downarrow$        | TR    | IR    | TR    | IR    | TR    | IR    | TR       | IR    | TR       | IR    | TR    | IR    |
> | ALBEF    | ETU [7] | 78.01 | 84.56 | 29.92 | 35.91 | 14.33 | 22.03 | 23.77    | 39.2  | 33.55    | 47.69 | 22.61 | 32.28 |
> |    ALBEF      | Ours       | **90.13** | **88.82** | **62.11** | **64.48** | **20.53** | **39.38** | **43.1**    | **65.93** | **54.4**     | **72.51** | **44.79** | **56.36** |
> | CLIP_VIT | ETU [7] | 14.8  | 25.23 | 21.22 | 30.87 | 10.87 | 24.96 | 84.14    | 90.45 | 57.51    | 65.51 | 16.4  | 27.22 |
> |   CLIP_VIT        | Ours       | **23.23** | **38.67** | **25.05**| **41.79** | **15.85** | **35.59** | **88.92**    | **93.05** | **66.06**    | **75.42** | **26.71** | **45.7**  |
>
>
> By contrastively training our designed cross-modal conditional generator, the proposed C-PGC greatly enhances the attack and achieves significant improvements in ASR. Particularly in the more realistic and challenging transferable scenarios, the proposed method achieves considerably better performance, *e.g.*, **32.19\% and 28.57\%** increase in ASR of TR and IR tasks when transferring from ALBEF to TCL. These results confirm the superiority of our contrastive learning-based generative paradigm. We will cite this paper and include these results in a new section of the Appendix.
>
> Thanks again for mentioning this great work! We would appreciate it if you could kindly inform us of any additional UAP methods targeting VLP models that we may have inadvertently omitted. We would be delighted to reproduce and compare these methods to ensure a comprehensive assessment.
>
> [1] Dong Y, Liao F, Pang T, et al. Boosting adversarial attacks with momentum. In CVPR, 2018.
>
> [2] Chakraborty A., Alam M., Dey V., et al. A survey on adversarial attacks and defences. In CAAI Transactions on Intelligence Technology, 2021.
>
> [3] Zhang J, Yi Q, Sang J. Towards adversarial attack on vision-language pre-training models. In ACM MM, 2022.
>
> [4] Lu D, Wang Z, Wang T, et al. Set-level guidance attack: Boosting adversarial transferability of vision-language pre-training models. In ICCV, 2023.
>
> [5] Wang H, Dong K, Zhu Z, et al. Transferable multimodal attack on vision-language pre-training models. In S&P, 2024.
>
> [6] Zhang T, Kishore V, Wu F, et al. Bertscore: Evaluating text generation with bert. In ICLR, 2020.
>
> [7] Zhang, Peng-Fei, Zi Huang, and Guangdong Bai. Universal Adversarial Perturbations for Vision-Language Pre-trained Models. In ACM SIGIR, 2024.

---

### Author Response · Authors · 2024-11-20
**Revision Uploaded**

We would like to express our sincere gratitude once more for your invaluable time and efforts in evaluating our work!

`A revision of our paper` that fully incorporates your precious suggestions along with the details concerning the revised content **has been uploaded**.
We have highlighted all modified content in the paper with blue color. Specifically, we have added/updated the following contents in the paper。
> **Reviewer jJKC**
1. We clarify the claim of *"Fully utilize the characteristics of V+L scenario"* and encourage future studies to investigate task-level characteristics in **Appendix I**.
2. We add the comparison with PGD-based UAP to **Figure 1** and cite the support literature in the **Introduction section**.
3. We highlight that our Appendix includes the cross-domain experiments in the **introduction part of Section 4**.
4. We provide the analysis regarding semantic similarity between adversarial text and its original text in **Appendix D**.
5. We conduct ablation studies of diverse target texts in **Appendix E**.
6. We supplement further discussions of unified losses in **Appendix B**.
7. We compare our C-PGC with the suggested baseline in **Appendix C**.
8. We compare different data augmentations in **Appendix E**.

> **Reviewer AoYQ**
1. We provide the analysis regarding semantic similarity between adversarial text and its original text in **Appendix D**.
2. We provide the way to calculate ASR in **Section 4.2**.
3. We discuss the employment of special characters in **Appendix J**.
4. We reorganize and highlight the innovations and contributions in the **contributions of Introduction**.

>  **Reviewer U4zG**
1. We provide the detailed motivation and rationale behind the contrastive loss in **Appendix B**.
2. We discuss the trade-offs between C-PGC and the suggested method in  **Appendix I**.
3. We compare our C-PGC with a recent SOTA baseline in **Appendix C**.
4. We improve the color schemes and visual materials of the framework diagram in  **Figure 3 of the Method**.

>  **Reviewer 4NRc**
1. We highlight the positive sample selection strategy in the **multimodal contrastive loss part of Method section**.
2. We discuss the effect of set-level augmentation in **Table 11 of Appendix E**.

---

### Meta-Review · Area_Chair_ngtZ · 2024-12-18

**Metareview:**

This work reveals that Vision-Language Pre-training (VLP) models are vulnerable to instance-agnostic Universal Adversarial Perturbations (UAPs). It introduces a Contrastive-training Perturbation Generator with Cross-modal conditions (C-PGC) that leverages malicious contrastive learning to disrupt the multimodal alignment in VLP models, achieving effective and transferable attacks across various models and vision-language tasks. The authors provided additional experiments and explanations during the rebuttal phase to address the reviewers' concerns; however, some issues remain unresolved. The primary weaknesses of this work lie in the inefficiency and ineffectiveness of the proposed universal adversarial text generation approach. The method's reliance on identifying critical words before applying universal perturbations contradicts the principle of universal attacks, which should function on all unseen data without additional computation. The quality of the generated perturbations is questionable, as merely replacing words does not ensure imperceptibility, making the modified texts easily identifiable. Additionally, the high semantic similarity between the original and adversarial texts suggests the attack has minimal impact, undermining its effectiveness. The use of large language models to validate imperceptibility is also unconvincing. Furthermore, the performance improvement primarily stems from the underlying perturbation technique (e.g., DeepFool) rather than the novel strategy proposed in this work, as highlighted in the rebuttal. Therefore, we decide not to accept this work based on its current state.

**Additional Comments On Reviewer Discussion:**

The authors provided additional experiments and explanations during the rebuttal phase to address the reviewers' concerns; however, some issues remain unresolved. The primary weaknesses of this work lie in the inefficiency and ineffectiveness of the proposed universal adversarial text generation approach. The quality of the generated perturbations is questionable, as merely replacing words does not ensure imperceptibility, making the modified texts easily identifiable. Additionally, the high semantic similarity between the original and adversarial texts suggests the attack has minimal impact, undermining its effectiveness. The use of large language models to validate imperceptibility is also unconvincing.

---

### Decision · Program_Chairs · 2025-01-22

Reject